# Recognition of an Ala-rich C-degron by the E3 ligase Pirh2

Xiaolu Wang[1,2,8], Yao Li [1,3,8], Xiaojie Yan [1,3,8], Qing Yang[3], Bing Zhang [3], Ying Zhang[4], Xinxin Yuan[3], Chenhao Jiang[4], Dongxing Chen [5], Quanyan Liu[6], Tong Liu[7], Wenyi Mi [1,4], Ying Yu [1,2] ✉ & Cheng Dong [1,3,6,7] ✉

The ribosome-associated quality-control (RQC) pathway degrades aberrant nascent polypeptides arising from ribosome stalling during translation. In mammals, the E3 ligase Pirh2 mediates the degradation of aberrant nascent polypeptides by targeting the C-terminal polyalanine degrons (polyAla/C-degrons). Here, we present the crystal structure of Pirh2 bound to the polyAla/C-degron, which shows that the N-terminal domain and the RING domain of Pirh2 form a narrow groove encapsulating the alanine residues of the polyAla/C-degron. Affinity measurements in vitro and global protein stability assays in cells further demonstrate that Pirh2 recognizes a C-terminal A/S-X-A-A motif for substrate degradation. Taken together, our study provides the molecular basis underlying polyAla/C-degron recognition by Pirh2 and expands the substrate recognition spectrum of Pirh2.

Protein homeostasis requires precise control of protein synthesis, folding and degradation, which is achieved by the coordinated action of translation machinery, molecular chaperones, the ubiquitin–proteasome system (UPS), and the autophagy machinery[1,2]. Deficiencies in this coordination have already been implicated in a variety of human diseases ranging from cancer and aging to neurodegenerative disorders[3–5]. To ensure proteome fidelity, cells have evolved a number of protein quality-control pathways that survey proteins to rapidly recognize and either correct or degrade aberrant proteins[6–8]. One such surveillance pathway is ribosome-associated quality control (RQC) that targets for degradation the potentially toxic nascent polypeptides produced by defective translation[9,10].

The RQC pathway is induced by ribosome stalling on mRNA during translational elongation, which may take place due to the defects in mRNA or ribosomes including truncated mRNA, inefficient

decoding and non-stop mRNA[11–14]. In eukaryotes, when RQC occurs, especially as the A site of the ribosome is empty such as on a truncated mRNA, the PELO-HBS1L complex together with ABCE1 (Dom34, Hbs1 and Rli1 in yeast) senses the stalled ribosome and separates the large 60 S and small 40 S ribosomal subunits[15–22]. During this process, the free 40 S subunit can be recycled, and the aberrant mRNA is degraded by exoribonuclease and the exosome complex[10,23], while the incomplete nascent chains remain attached to the 60S subunit, and thus awaiting further processing via the RQC-L or RQC-C pathway[24,25]. In addition, the RQC-Trigger (RQT) complex is a major dissociation factor to dissociate the collided ribosomes. The RQT complex recognizes ubiquitinated collided ribosomes as a substrate to initiate the RQC pathway by ribosomal subunit dissociation[22,26–28].

In the RQC-L pathway, NEMF (nuclear export mediator factor, Rqc2 in yeast) recognizes the obstructed large subunit and recruits

[1]The Province and Ministry Co-sponsored Collaborative Innovation Center for Medical Epigenetics, Key Laboratory of Immune Microenvironment and Disease (Ministry of Education), Haihe Laboratory of Cell Ecosystem, School of Basic Medical Sciences, Tianjin Medical University, 300070 Tianjin, China. [2]Department of Pharmacology, Tianjin Key Laboratory of Inflammatory Biology, Center for Cardiovascular Diseases, Tianjin Medical University, 300070 Tianjin, China. [3]Department of Biochemistry and Molecular Biology, Tianjin Medical University, 300070 Tianjin, China. [4]Department of Immunology, Tianjin Institute of Immunology, Tianjin Medical University, 300070 Tianjin, China. [5]Department of Medicinal Chemistry, Tianjin Key Laboratory on Technologies Enabling Development of Clinical Therapeutics and Diagnostics, School of Pharmacy, Tianjin Medical University, 300070 Tianjin, China. [6]Department of Hepatobiliary Surgery, Tianjin Medical University General Hospital, 300052 Tianjin, China. [7]Department of Cardiology, Tianjin Institute of Cardiology, Second Hospital of Tianjin Medical University, 300211 Tianjin, China. [8]These authors contributed equally: Xiaolu Wang, Yao Li, Xiaojie Yan. ✉e-mail: yuying@tmu.edu.cn; dongcheng@tmu.edu.cn

Listerin E3 ligase to target nascent chains for proteasomal degradation[29,30]. In addition, NEMF can recruit tRNA-Ala to elongate the nascent chain with C-terminal Ala tail (Ala and Thr tail, CAT tail in yeast), which facilitates ubiquitination of the incomplete nascent chain by Listerin E3 ligase through exposing lysine buried in the ribosomal exit tunnel[24,31,32].

If the Listerin activity is limited under certain conditions, the aberrant nascent chain could be proteolyzed by the RQC-C pathway. In RQC-C, the NEMF-mediated C-terminal Ala tail serves as a C-terminal degradation signal (C-degron) that can be recognized and ubiquitinated by the E3 ligase Pirh2 or KLHDC10 for degradation through the C-degron pathway after release from the 60S subunit[24,25]. In addition, the C-degron pathway is also present in bacteria despite the absence of the UPS. Instead, the SmrB-tmRNA (transfer-messenger RNA)-mediated C-terminal SsrA tag or RqcH-mediated C-terminal Ala tail is recognized for degradation by the proteasome-like ClpXP protease[33–36], suggesting that the C-degron pathway plays an important role in protein quality control.

The eukaryotic C-degron pathway, analogous to the N-degron pathway, has recently been identified as a new proteolytic pathway, in which a specific set of Cullin-RING E3 ubiquitin ligases (CRLs) act as C-recognins that recognize distinct C-degrons for degradation[37–39]. Specifically, the KLHDC10 has been shown to target the Gly/C-degron for degradation[37,40]. KLHDC10 is supposed to utilize a common Kelch repeat β-propeller fold, as in KLHDC2, to recognize the substrate's C-degron[41].

Of note, Pirh2 has previously been characterized as an E3 ligase that promotes p53 degradation in response to DNA damage[42,43]. Its dysregulation has been associated with various types of cancer, thereby being a potential cancer therapeutic target[44–46]. Besides, the non-CRL E3 ligase Pirh2, as a novel C-recognin, not only recognizes the aberrant protein, but can also target native proteins ending with an Ala-rich C-degron for degradation[24]. However, the substrate specificity and recognition mechanism by Pirh2 remain elusive. Here, we provide the crystal structure of Pirh2 in complex with a C-terminal polyalanine degron (polyAla/C-degron) and reveal the molecular mechanism underlying Pirh2-mediated recognition of polyAla/C-degron in the C-degron pathway. Furthermore, we investigate the substrate specificity and identify new degrons targeted for degradation by Pirh2. Our study not only expands the range of substrate recognition by Pirh2, but also lays the foundation for future development of chemical probes.

## Results

### The NTD and RING domains of Pirh2 are important for Ala6/C-degron binding

The E3 ligase Pirh2 (p53-induced RING-H2 protein), also known as Rchy1 (RING finger and CHY zinc finger domain-containing protein 1), encompasses nine zinc binding sites throughout the full-length protein, with six in the N-terminal domain (NTD or CHY zinc finger), two in the central RING domain and one in the C-terminal domain (CTD) (Fig. 1a). A recent study showed that Pirh2 can bind to a C-terminal six-Ala degron (Ala6/C-degron) in the RQC pathway[24]. However, it is yet unclear exactly which domain(s) of Pirh2 are responsible for the interaction with the Ala6/C-degron, although the NTD and CTD were reported to mediate the recognition of the native substrate p53, which lacks a C-terminal Ala-rich tail[42]. To this end, we purified a series of Pirh2 segments covering various domains and performed glutathione S-transferase (GST) pull-down assays using GST-fused Ala6/C-degron as bait. The results showed that the NTD and NTD + RING, like the

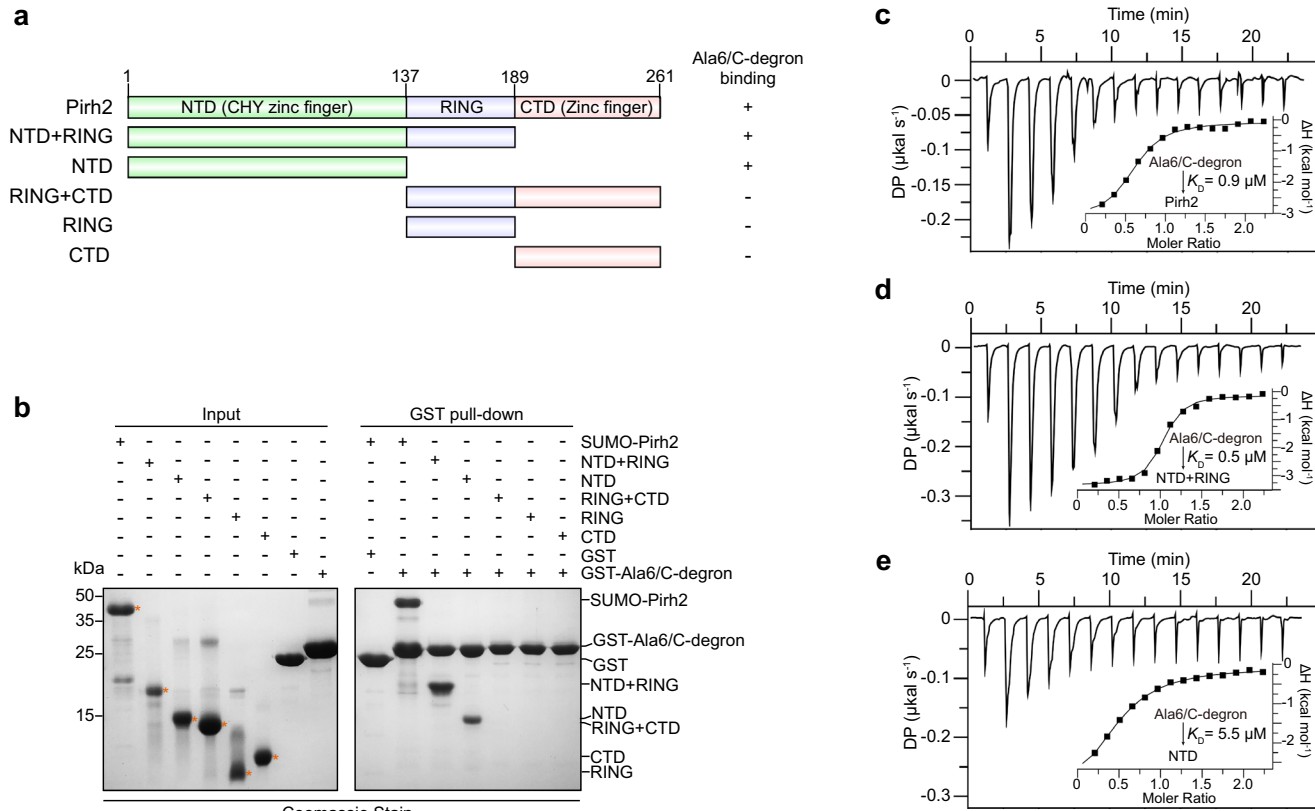

**Fig. 1 | The NTD and RING of Pirh2 coordinate in Ala6/C-degron binding.**
**a** Domain architecture of human Pirh2 and fragments of Pirh2 used in the GST pull-down experiment. NTD, N-terminal domain; CTD, C-terminal domain. RING, really interesting new gene. **b** GST pull-down assay using GST-fused Ala6/C-degron to pull down purified full-length and different Pirh2 fragments. Representative images, $n = 3$. Source data are provided as a Source Data file. **c–e** ITC titration and fitting curves of full-length Pirh2, NTD + RING, and NTD with Ala6/C-degron peptide (sequence: YKTSAAAAAA). Binding affinities ($K_D$) are indicated.

**Table 1 | Data collection and refinement statistics**

| | Pirh2-polyAla |
|---|---|
| PDB accession number | 7YNX |
| **Data collection** | |
| Space group | P 2$_1$2$_1$2$_1$ |
| Cell dimensions | |
| a, b, c (Å) | 69.47, 80.27, 84.87 |
| α, β, γ (°) | 90.00, 90.00, 90.00 |
| Resolution (Å) | 37.52–2.30 (2.38–2.30)[a] |
| R$_{sym}$ or R$_{merge}$ | 0.247 (1.328) |
| I/σI | 8.62 (2.22) |
| Completeness (%) | 90.6 (81.1) |
| Redundancy | 12.7 (12.5) |
| **Refinement** | |
| Resolution (Å) | 37.52–2.30 (2.38–2.30) |
| No. reflections | 19634 (1721) |
| R$_{work}$/R$_{free}$ | 0.2159/0.2644 |
| **No. of atoms** | |
| Protein | 2985 |
| Water | 155 |
| Ligand/ion | 53 |
| **B-factors** | |
| Protein | 31.9 |
| Water | 35.1 |
| Ligand/ion | 55.1 |
| **R.m.s. deviations** | |
| Bond lengths (Å) | 0.007 |
| Bond angles (°) | 0.91 |

[a]Values in parentheses are for highest-resolution shell.

full-length Pirh2, are able to bind Ala6/C-degron (Fig. 1b). In contrast, the individual RING, CTD and RING + CTD that lacking the NTD, failed to pull down the Ala6/C-degron (Fig. 1b), suggesting that the NTD is necessary for Ala6/C-degron binding.

To further quantify this interaction, we synthesized a decapeptide containing the Ala6/C-degron (sequence: YKTSAAAAAA) and carried out isothermal titration calorimetry (ITC) measurements. Consistent with GST pull-down assays, the full-length Pirh2, NTD + RING and NTD are capable of binding directly to the Ala6/C-degron peptide (Fig. 1c–e). Nevertheless, our results clearly showed that the NTD + RING exhibited a robust interaction with the Ala6/C-degron with a $K_D$ value of 0.5 μM (Fig. 1d), which is comparable to that of the full-length Pirh2 ($K_D$ of 0.9 μM) (Fig. 1c). Whereas the single NTD displayed an 11-fold-decreased binding affinity ($K_D$ of 5.5 μM) compared to NTD + RING (Fig. 1e). These observations indicate that the RING makes a substantial contribution to the substrate recognition, although it cannot efficiently mediate this interaction alone. Taken together, our results demonstrate that the Pirh2 NTD + RING is sufficient for Ala6/C-degron recognition, with the NTD being required, the RING being auxiliary, and the CTD being unnecessary for this interaction.

### Crystal structure of Pirh2 bound to the Ala6/C-degron

To better understand the recognition mode of the Ala6/C-degron by Pirh2, we determined the crystal structure of the Pirh2 NTD + RING domains bound to the Ala6/C-degron by fusion of a linker and a six-Ala peptide at the C-terminus. Data collection and structure refinement statistics are summarized in Table 1. As expected, the Pirh2 NTD and RING coordinate eight zinc ions, whereby each zinc ion is coordinated by four Cys/His residues (Fig. 2a). Interestingly, the overall structure of NTD + RING forms a π-shaped conformation, with the polyAla tail

inserted parallel to the bottom of the width axis (Fig. 2a and Supplementary Fig. 1). In light of the complex structure, we found that the Pirh2-mediated Ala6/C-degron binding pocket exhibits several unique features:

1. this entire binding pocket is formed by the NTD and RING together, whereas the CTD is predicted to be located far away from the Ala6/C-degron binding pocket (Fig. 2b), thereby making a minor contribution to the substrate recognition as supported by our GST pull-down and ITC results. Remarkably, the Pirh2 RING domain is directly involved in Ala6/C-degron binding (Fig. 2a, b), although other E3 ligase RING domains are generally thought to mediate the binding of ubiquitin-conjugating enzymes (E2s) and promote the transfer of ubiquitin to the substrate[47]. In addition, the Ala6/C-degron, especially the extreme C-terminal Ala (Ala-1), is predominantly docked onto the NTD contacts, and the upstream residues of Ala-1 are further buttressed by the RING domain (Fig. 2a, b). These observations help to explain why the NTD is necessary for the Ala6/C-degron binding, and why the RING is found to assist in this binding.

2. this binding pocket has a positively charged path (Fig. 2c). The carboxyl group of Ala-1, like other C-degrons, is coordinated in a basic pocket[41,48]. Aside from that, this binding pocket is evolutionarily conserved in sequence across different species (Fig. 2d). In particular, the region involved in the Ala-1 to Ala-4 binding is highly conserved (Fig. 2d), implying a conserved polyAla/C-degron recognition mode.

3. this binding pocket presents a linear and long (~30 Å) deep groove, with an open and a closed end (Fig. 2b). Moreover, the downstream groove spanning Ala-1 to Ala-4 bound is narrower than the upstream region (Fig. 2e). The binding interface with the six-Ala peptide buries a total of 528 Å$^2$ solvent-accessible surface area, in which the backbone of the Ala6/C-degron lies exactly at the base of the binding groove with the methyl groups oriented in distinctive directions (Fig. 2d, e). Specifically, the methyl groups of Ala-1, Ala-2, Ala-4, and Ala-5 each point toward a closed cavity inside the groove walls, while the methyl groups of Ala-3 and Ala-6 point toward the groove opening (Fig. 2e). In this long groove, a number of Ala residues, at least six, could be accommodated.

We speculate that Pirh2, like other N- or C-recognins, may mainly bind to a certain length of C-degron[48–53]. To map the minimal length of the polyAla/C-degron for Pirh2 binding, we examined the binding affinities of Pirh2 with polyAla/C-degron peptides of varying Ala residues. Our ITC results showed that the peptide containing four to six Ala residues retains binding activities, while the three-Ala or two-Ala peptide resulted in significant loss of Pirh2 binding (Fig. 2f). Therefore, we conclude that the minimal four-Ala degron is sufficient to bind Pirh2, which is in agreement with previous data that at least four-Ala degron can be efficiently degraded by Pirh2[24].

### Molecular mechanism of Ala6/C-degron recognition

Pirh2 recognizes the Ala6/C-degron through a combination of hydrophobic and hydrophilic contacts. Concretely, the free carboxyl group of Ala-1 is positioned to form two hydrogen bonds with the hydroxyl group of Tyr23 and one salt bridge with Arg41 (Fig. 3a, g). Simultaneously, the carboxyl group of Ala-1 is further stabilized by water-mediated hydrogen bonds with Arg54, Asp94 and Gln99 (Fig. 3a, g). The backbone amide group of Ala-1 forms a direct hydrogen bond with the backbone carbonyl group of Cys108 coordinating the fourth zinc (Fig. 3a, g). Furthermore, the methyl group of Ala-1 is flanked by the hydrophobic side chains of Leu92 and Ile110 (Fig. 3a). As a result, the Ala-1 is locked firmly at the closed end of the groove. Notably, these Ala-1 interacting residues are mainly contributed by the Pirh2 NTD (Figs. 2a and 3a), it is thus not surprising that the NTD is necessary for Ala6/C-degron binding as shown by our GST pull-down assay (Fig. 1b).

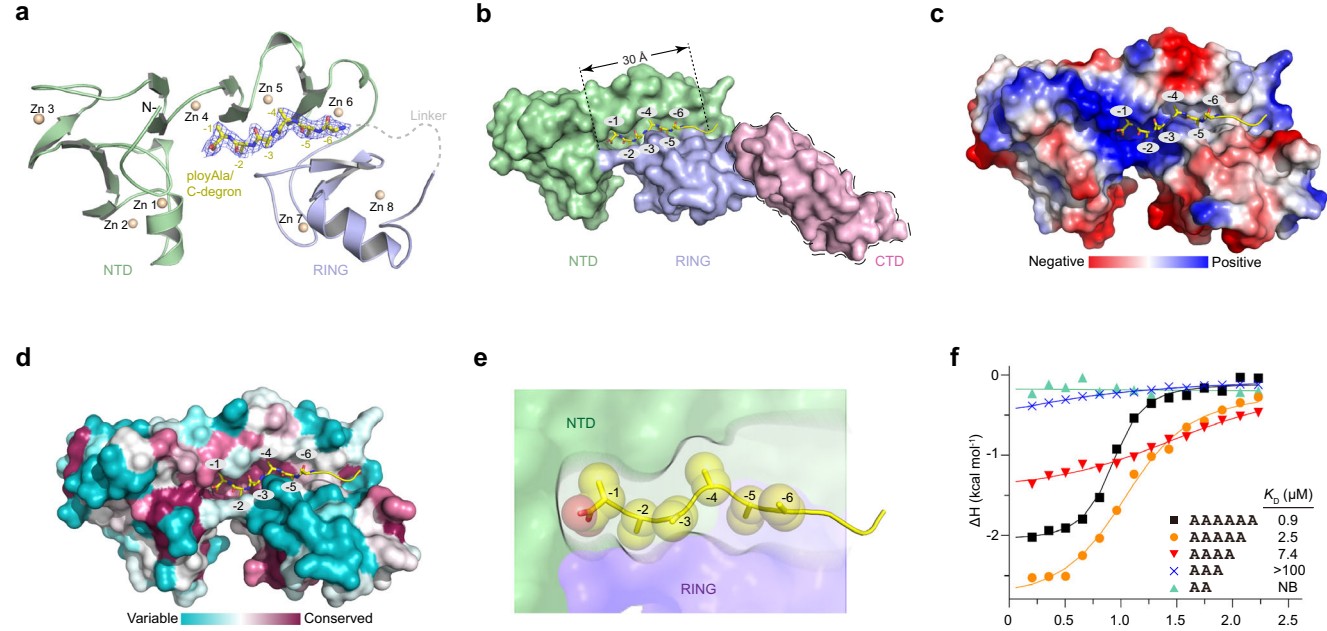

**Fig. 2 | Overall structure of Pirh2 bound to the Ala6/C-degron. a** Ribbon diagram of Pirh2 in complex with the Ala6/C-degron. NTD and RING subdomains are colored with palegreen and lightblue, respectively. Ala6/C-degron is colored with yellow. 2Fo−Fc electron density map of the six-Ala is shown in blue and contoured to 1.0σ. Eight zinc ions are shown as lightorange spheres. **b** The surface view of Pirh2 (NTD + RING) in complex with the Ala6/C-degron. The location of CTD is predicted by AlphaFold v2.0 (https://www.alphafold.ebi.ac.uk/). NTD, RING and CTD (surrounded by dotted line) subdomains are colored with palegreen, lightblue and lightpink. Ala6/C-degron is colored with yellow. **c** The electrostatic potential surface of the Ala6/C-degron-binding pocket in Pirh2 (red, negative; blue, positive).

**d** Conservation analysis of Pirh2 (NTD + RING) using the ConSurf server (https://consurf.tau.ac.il/). The residues of Pirh2 (NTD + RING) are colored based on their conservation grades using the color-coding bar, with turquoise-through-maroon indicating variable-through-conserved residues. **e** Schematic representation of the Ala6/C-degron binding groove in Pirh2 (NTD + RING). Ala6/C-degron is shown as spheres. **f** ITC fitting curves of Pirh2 (NTD + RING) with variable lengths of Ala6/C-degron peptides. The corresponding peptide lengths, sequences and binding affinities ($K_D$) are indicated. NB, no detectable binding under our experimental conditions.

The Ala-2 is accommodated in a narrow pocket created by NTD and RING, where a backbone hydrogen bond is formed between the amide group of Ala-2 and the carbonyl oxygen of Ser155, and the side chain of Ile107 establishes a hydrophobic interaction with the methyl group of Ala-2 (Fig. 3b, g).

Unlike Ala-1 and Ala-2, the methyl group of Ala-3 expands toward the opening of the groove, and is embraced by hydrophobic clusters formed by Ile110 in NTD, Val157 and Val158 in RING (Fig. 3c). Meanwhile, the backbone of Ala-3 is further anchored by the backbone of Ile110 through a direct and a water-mediated hydrogen bond (Fig. 3c, g).

The methyl group of Ala-4 is located in a shallow cavity, neighboring the fifth zinc binding site, generated by Tyr100, Arg109, Phe116 and Cys125 in NTD (Fig. 2a, d). The backbone carbonyl and amide groups form two direct hydrogen bonds with the guanidine group of Arg109 and the backbone carbonyl group of Cys125, respectively (Fig. 3d, g). Besides, these contacts are further bridged by two waters at the base of the groove (Fig. 3d, g).

The methyl group of Ala-5 is inserted into a hydrophobic pocket formed by Leu124, Leu167 and Ala159 (Fig. 3e). The backbone amide group of Ala-5 further hydrogen bonds with the backbone carbonyl group of Ala159 in the RING (Fig. 3e, g).

The Ala-6 is located at the open end of the groove (Figs. 2e and 3f). Similar to the Ala-3, the methyl group of Ala-6 faces the groove opening, and is packed by hydrophobic Val158 and Ala127 (Fig. 3f). The backbone forms a direct and a water-mediated hydrogen bond with Ala127 and Asn129, respectively (Fig. 3f, g).

Overall, Pirh2 utilizes the NTD and RING domains forming a long groove to engage the Ala6/C-degron. Ala-1 and Ala-6 are accommodated at the closed and open ends of the groove, respectively. The

backbones of the Ala6/C-degron are coordinated by a network of direct and water-mediated hydrogen bonds, and the methyl groups of the Ala6/C-degron are further stabilized through hydrophobic residues lining the groove.

## Key residues of Pirh2 for Ala6/C-degron binding and substrate degradation

To substantiate the key residues of Pirh2 in mediating Ala6/C-degron binding, we generated point mutations of Pirh2 and examined their binding affinities toward Ala6/C-degron peptide by ITC. Indeed, these mutants reduced the substrate binding capacity in vitro. For example, the mutants of Y23A and R41A, which would abolish the hydrogen bonds with the carboxyl group of Ala-1 (Fig. 3a), resulted in complete loss of substrate binding (Fig. 4a), highlighting the importance of these hydrogen bonds in mediating Ala-1 recognition. Moreover, the mutants on residues bridging the water-mediated contacts, such as R54A, D94A, or Q99A (Fig. 3a), diminished in binding affinity by more than 6-fold (Fig. 4a). Substitutions of Leu92 and Ile110 with negatively charged residues severely impaired the substrate binding (Fig. 4a), suggesting a critical role of hydrophobic interactions in Ala-1 binding. In addition, the R109A mutant totally disrupted the substrate binding (Fig. 4a), which is in line with our structural analysis that Arg109 braces the backbone of Ala-4 via direct and water-mediated hydrogen bonds (Fig. 3d). On the other hand, substitutions of Ser155 and Ala159 in the RING domain with aspartic acid residues decreased the binding affinity by 9- and 14-fold, respectively, suggesting that the RING domain also plays an important role in the Ala6/C-degron binding (Fig. 4a).

To further corroborate the functional significance of the key residues directly interacting with Ala6/C-degron in cells, we generated a global protein stability (GPS) reporter cell line[37,54] using a bicistronic

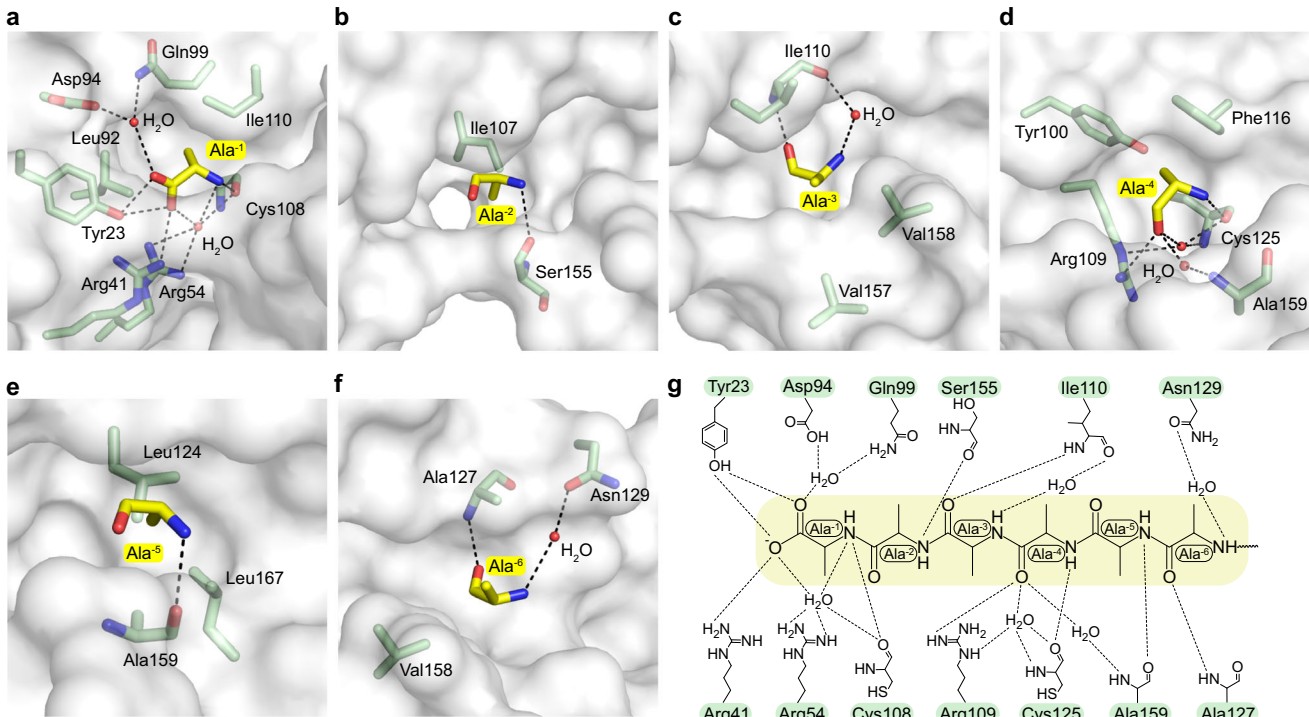

**Fig. 3 | Molecular basis of Ala6/C-degron recognition by Pirh2. a–f** Close-up view of the interactions of Pirh2 with Ala-1 to Ala-6 in the binding groove. Alanine residues of Ala6/C-degron are shown as yellow sticks; residues of Pirh2 that are involved in the interactions are shown as green sticks. The water molecules are shown as red spheres. The hydrogen bonds are shown as black dashed lines. **g** A schematic illustration of the interaction of Ala6/C-degron with Pirh2.

reporter construct encoding a DsRed, which serves as an internal reference, and a green fluorescent protein (GFP)-fused Ala6/C-degron, which is translated from an internal ribosome entry site (IRES) (Fig. 4b). We evaluated the protein stability by quantification of the GFP/DsRed ratio since the DsRed and GFP fusion protein are expressed from the same transcript. Immunoblot initially revealed that Pirh2 knockdown by shRNA (small hairpin RNA) markedly inhibited the degradation of GFP-fused Ala6/C-degron rather than the internal control DsRed (Supplementary Fig. 2a, b). Then, GPS assay also showed that knockdown of endogenous Pirh2 stabilized the GFP-fused Ala6/C-degron displaying an increased GFP/DsRed ratio, while overexpression of wild-type (WT) Pirh2 rescued the degradation of GFP-fused Ala6/C-degron protein with a decreased GFP/DsRed ratio (Fig. 4c, d), indicating that Ala6/C-degron-mediated GFP degradation is dependent on Pirh2. In contrast to WT Pirh2, overexpression of the mutants (including Y23A, L92D, and R109A), which disrupted Pirh2 binding ability to Ala6/C-degron, failed to promote the degradation of GFP-fused Ala6/C-degron (Fig. 4c, d), further confirming the essential role of these Ala6/C-degron interacting residues in the substrate degradation.

### Substrate specificity of Pirh2-mediated Ala6/C-degron recognition

It has been well established that Pirh2 targets the aberrant proteins ending with polyAla/C-degron for degradation[24]. On the other hand, some data indicated that Pirh2 may recognize native substrates harboring Ala-rich C-degrons through the same binding pocket[24]. To identify the substrate specificity of Pirh2, we first sought to map the key elements of the Ala6/C-degron in Pirh2 recognition. Hence, we performed threonine scanning mutagenesis and GST pull-down assays. As shown in Fig. 5a, the threonine mutations of Ala-1, Ala-2 and Ala-4 defected in the Pirh2 NTD + RING binding. In contrast, the threonine mutations of Ala-3, Ala-5 and Ala-6 retained their binding capacity to Pirh2 (Fig. 5a). Next, we further examined their binding affinities by ITC. We found that the threonine mutations of Ala-1 and Ala-4 did

indeed abrogate Pirh2 binding (Fig. 5b). The threonine mutations of Ala-2 dramatically decreased the binding affinity by 30-fold, whereas the threonine mutation of Ala-3 showed a slight reduction in binding affinity compared to the Ala6/C-degron peptide (Fig. 5b), suggesting that the −1, −2, and −4 positions other than the −3 position are critical for Pirh2 binding. However, the threonine mutations of Ala-5 and Ala-6 exhibited almost the same binding affinities as the Ala6/C-degron (Fig. 5b), further reinforcing our above result that four-Ala degron is sufficient for Pirh2 binding.

To extend these observations, we examined the effects of Pirh2 on the stability of GFP-fused Ala6/C-degron with threonine substitutions in cells in which KLHDC10, another E3 ligase that can act redundantly with Pirh2 in RQC, had been knocked down by shRNA. Our GPS assays showed that the threonine substitutions of Ala-1, Ala-2 and Ala-4 led to a major stabilization of GFP, whereas the substitutions of Ala-3, Ala-5 and Ala-6 had almost the same behavior as the wild-type Ala6/C-degron (Supplementary Fig. 3), which is in agreement with our in vitro results.

To further explore the substrate preference ranging from −1 to −4 positions, we synthesized a series of peptides with −1 to −4 substitutions and tested their binding affinities for Pirh2 NTD + RING. Our ITC results showed that replacement of Ala-1 with the smallest glycine residue caused a ~29-fold decrease in binding affinity, while mutating Ala-1 to any other small residue, such as serine, valine or proline, completely abolished the Pirh2 binding (Fig. 5c). On the other hand, since Ala-1 sits at the closed end of the basic groove (Fig. 2b, c, e), neither the addition of an extra threonine following Ala-1 nor the amidation of the Ala-1 was able to bind Pirh2 (Fig. 5c). Together, these data demonstrate that Pirh2 specifically recognizes the free C-terminal alanine within the Ala6/C-degron through a limit and basic pocket.

Substitution of Ala-2 with small residues or hydrophobic isoleucine displayed a significant decrease in binding affinities, ranging from 16- to 32-fold (Fig. 5d). Mutating Ala-2 to the bulky hydrophobic

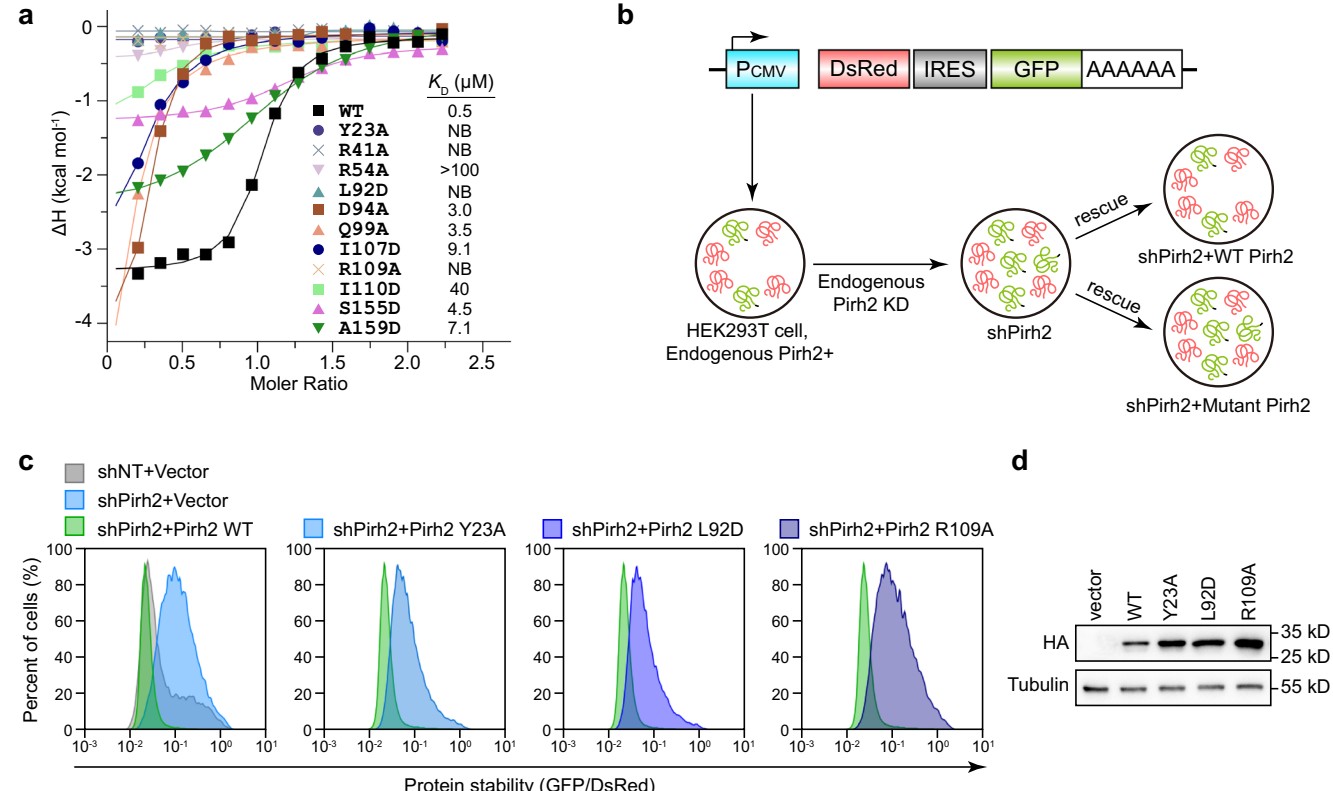

**Fig. 4 | Mutagenesis analysis of Pirh2 for Ala6/C-degron binding. a** ITC fitting curves of wild-type (WT) and mutant Pirh2 titrated with the Ala6/C-degron peptide. The corresponding mutant proteins and binding affinities ($K_D$) are indicated. NB, no detectable binding under our experimental conditions. **b** Schematic representation of the GPS assay. $P_{CMV}$, cytomegalovirus promoter; DsRed, *Discosoma* sp. red fluorescent protein; IERS, internal ribosome entry site; GFP, green fluorescent protein. **c** Stability analysis of GFP-fused Ala6/C-degron in endogenous Pirh2 knockdown cells, and rescued by exogenously expressed WT and mutant Pirh2. The shRNA targets the 3′-untranslated region (UTR) of endogenous Pirh2. The ratio of GFP/DsRed was analyzed by flow cytometry. **d** Western blot analysis of HA-tagged WT and mutant Pirh2 expression in Ala6/C-degron GPS-reporter cell lines. Representative images, $n = 3$. Source data are provided as a Source Data file. All FACS sequential gating images are provided in Supplementary Fig. 9.

residue (Phe) or charged residue (Lys or Asp) led to a complete loss of binding (Fig. 5d). Therefore, Pirh2 favors an alanine at the position −2.

Surprisingly, the substitutions of Ala-3 only mildly affected the binding abilities, and that the proline substitution has a comparable binding affinity to the alanine (Fig. 5e). Recalling the binding mode, the methyl group of Ala-3 points toward the groove opening (Fig. 2e), where has enough space to accommodate other residues. Thus, we suggest that Pirh2 can tolerate a variety of residues with a preferred alanine and proline at position −3.

Intriguingly, substitution of Ala-4 with the small residue serine, but not glycine or proline, slightly weakened the binding affinity by ~6-fold (Fig. 5f). Replacement of Ala-4 with other residues such as Ile, Val or Lys failed to bind Pirh2 (Fig. 5f). So a small serine at the position −4 appears to be allowed for Pirh2 binding.

In sum, we conclude that Pirh2 prefers an Ala-rich C-degron, and that the last four residue confer the substrate specificity. Based on the specificity profile obtained from ITC assays, we suggest that Pirh2 can recognize substrates ending with an A/S-X-A-A motif, in which the serine is a suboptimal residue at position −4, and diverse types of residues can be tolerated at position −3.

### Pirh2-mediated substrate degradation in cells

To further verify the substrate specificity of Pirh2 in cells, we carried out GPS assays by constructing a single residue substitution in Ala6/C-degron with an otherwise identical counterpart. The above data indicated that Pirh2 preferentially recognizes a small alanine and disfavors other residues at the position −2 in vitro (Fig. 5d). In accordance with the in vitro results (Supplementary Fig. 4a), the serine, valine or bulky

isoleucine substitution of Ala-2 remarkably inhibited the degradation of the GFP fusion protein relative to Ala-2 in cells (Fig. 6a). Further, loss of Pirh2 did not stabilize these GFP-peptide fusion proteins due to the impaired Pirh2 binding (Fig. 6b). This observation is similar to a previous study showing that threonine substitution of Ala-2 resulted in stabilization of the GFP fusion protein, although it has a weak interaction with Pirh2 ($K_D = 15 \mu M$) by our ITC result (Fig. 5b). Therefore, the Ala-2 of polyAla/C-degron plays a vital role in the Pirh2-mediated degradation.

Importantly, we found that substitutions of Ala-3 with proline, isoleucine or lysine also readily interact with Pirh2 in our ITC assays (Fig. 5e and Supplementary Fig. 4b). Consistently, these substitutions of Ala-3 had an instability profile similar to the Ala6/C-degron in cells (Fig. 6c), and that these substitutions were responsively stabilized upon Pirh2 knockdown (Fig. 6d). Moreover, rescue experiments showed that the overexpression of exogenous WT Pirh2 but not the mutants Y23A, L92D or R109A permitted the degradation of the isoleucine substitution of Ala-3 (Fig. 6e, f), representing that the antepenultimate residue of the polyAla/C-degron exhibits more flexibility in Pirh2-mediated recognition and degradation.

Our ITC data have established that Pirh2 can recognize a C-terminal A/S-X-A-A motif, in which the serine is a suboptimal residue at position −4. As expected, knockdown of Pirh2 stabilized the serine substitution at position −4 (Supplementary Fig. 5). Correspondingly, overexpression of wild-type (WT) Pirh2 but not Y23A, L92D and R109A mutants can stimulate the degradation of S-A-A-A degron (Fig. 6g, h), indicating serine can be allowed at the position −4 in Pirh2-mediated degradation (Fig. 5f).

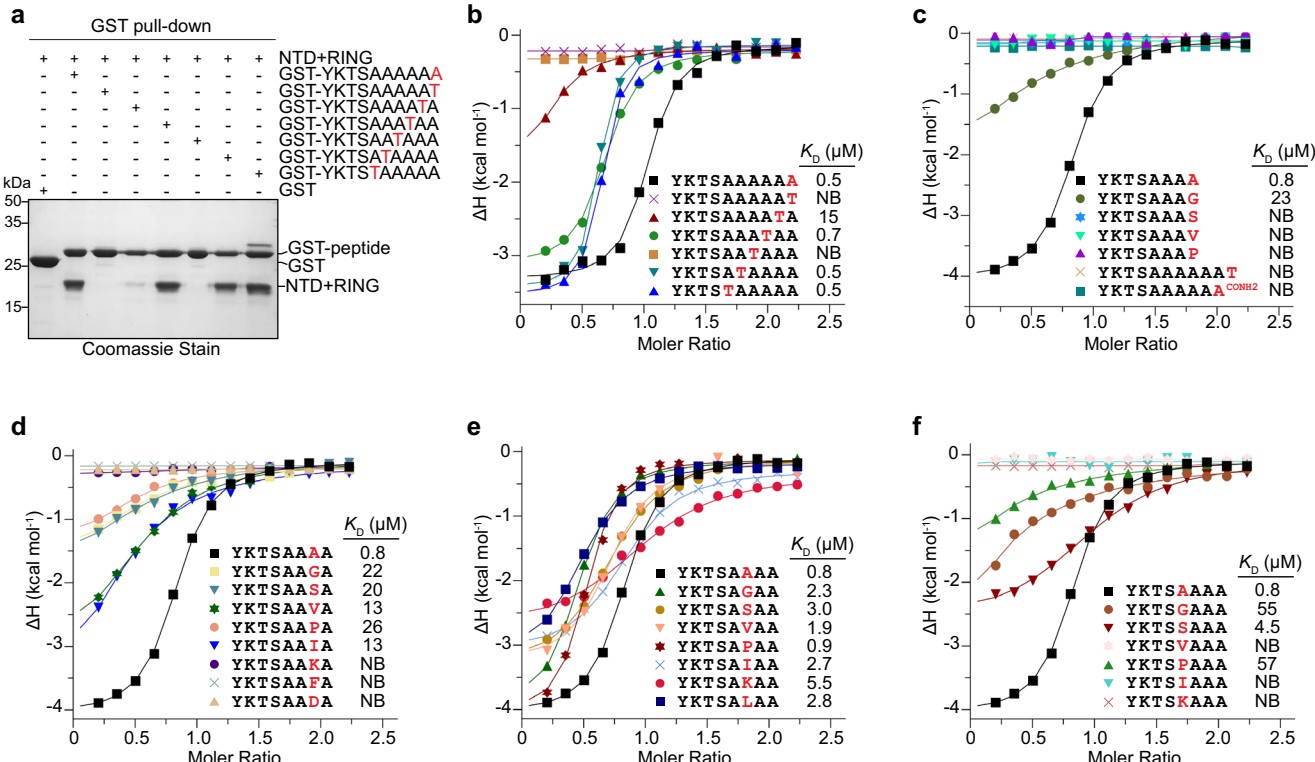

**Fig. 5 | Substrate specificity of Pirh2-mediated Ala6/C-degron recognition.**
**a** GST pull-down assay using GST-fused Ala6/C-degron or its mutants to pull down Pirh2 (NTD + RING). Representative images, $n = 3$. Source data are provided as a Source Data file. **b**–**f** ITC fitting curves of Pirh2 (NTD + RING) titrated by a series of peptides with position substitutions. The corresponding peptide sequences and binding affinities ($K_D$) are indicated. NB, no detectable binding under our experimental conditions.

To test whether Pirh2 is able to target native substrates ending in the A/S-X-A-A motif for degradation. We found that the ITPA (inosine triphosphate pyrophosphatase) protein, which ends in SLAA (-LQEYFGSLAA), is a potential substrate of Pirh2. Indeed, Myc-fused ITPA levels were increased upon Pirh2 knockdown by shRNA (Supplementary Fig. 6a). Furthermore, overexpression of wild-type Pirh2, but not the C-degron-binding-deficient mutants including Y23A, L92D or R109A, caused a reduction in ITPA levels (Supplementary Fig. 6b).

Therefore, based on all of the collective data, we conclude that Pirh2 can target non-strict polyAla/C-degrons for degradation. In particular, the degrons displayed much more flexibility at the −3 position, albeit with a limited residue such as serine at the −4 position.

## Discussion

Ribosomal stalling during translation is an accidental event that can occur for various reasons ranging from bacteria to humans[55]. The resultant incomplete products would be extremely toxic to cells. To prevent toxic accumulation, cells have to rescue stalled ribosomes, and eliminate the incomplete nascent polypeptides via the RQC pathway[25]. In the mammalian RQC pathway, the E3 ligase Pirh2 can directly recognize the polyAla-tailed ribosome stalling products, and target them for degradation through a new polyAla/C-degron pathway[24]. Herein, we decipher the molecular mechanism of Pirh2-mediated recognition of polyAla/C-degron by means of structural biology and biochemistry. We found that Pirh2 recognizes the polyAla/C-degron through both the NTD and the RING domains. Furthermore, Pirh2 can recognize and target the A/S-X-A-A C-degron for degradation.

In addition to Pirh2, the ployAla/C-degron is also sensed by the E3 ligase KLHDC10, which contains a Kelch repeat domain for substrate binding. We predicted the binding mode of KLHDC10 to the six-Ala tail by means of the structure predicted by Alphafold and molecular docking. The modeled ployAla/C-degron is embedded in a positively charged pocket, similar to that of other typical C-degron binding modes[41,48,53], and is shown here (Supplementary Fig. 7). Noteworthy, KLHDC10 utilizes a deep and narrow pocket to engage the six-Ala tail with a solenoidal conformation similar to the reported Gly/C-degron-bound KLHDC2 structure[41]. However, Pirh2 uses a long groove to recognize the linear six-Ala tail. Therefore, Pirh2 and KLHDC10 may tend to recognize their dedicated polyAla tail in different conformations.

Pirh2 is involved in a plethora of cellular processes through regulation of a number of factors[45]. Particularly, it is well documented that Pirh2 is a negative regulator of p53 by promoting p53 ubiquitination and degradation[43]. Structural and biochemical analyses have indicated that the physical interaction of the Pirh2 CTD with the p53 TET domain is essential for Pirh2-mediated p53 ubiquitination[42]. The NTD, on the other hand, just seems to augment Pirh2-mediated p53 ubiquitination through its weak interaction with the p53 DNA binding domain[42]. In the present study, we found that Pirh2 recognizes the Ala6/C-degron mainly by NTD, with the RING domain largely enhancing the binding. In contrast, the CTD does not seem to be responsible for the Ala6/C-degron recognition. Thus, it is likely that Pirh2 alternatively employs CTD or NTD as the main scaffold for specific substrate recognition. Of note, the scanning of the reported substrates of Pirh2, including c-Myc[56], HuR[57], p27[58], HDAC1[59], and so on, showed neither striking sequence identity nor enrichment of polyAla tails. We thus anticipate that the accurate substrate recognition mode by Pirh2 must be considerably complex.

Early studies suggested that the Pirh2 RING domain, like other E3 ligase RING domains, mainly recruits the ubiquitin-conjugating enzyme E2 (UBE2D2) and stimulates substrate ubiquitination[42,60], and is not predicted to recognize substrates. Here, we demonstrate that the RING domain of Pirh2, apart from its intrinsic functions, directly

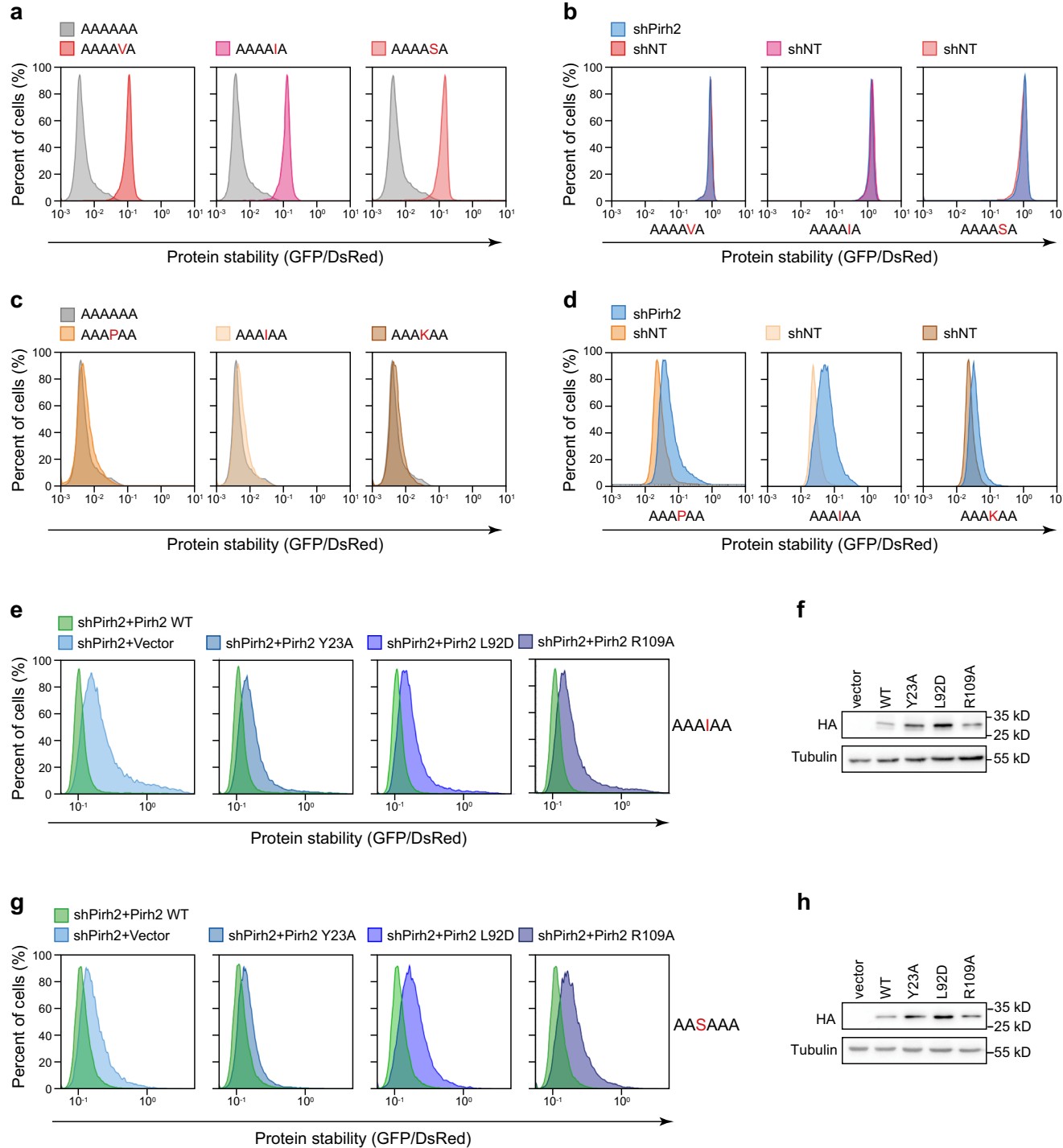

**Fig. 6 | Pirh2-mediated substrate degradation in cells. a** Stability comparison of GFP-fused Ala6/C-degron with Ala-2 substitutions in HEK293T cells, monitored by global protein stability assay. The ratio of GFP/DsRed was analyzed by flow cytometry. **b** Stability analysis of GFP-fused Ala-2 substitution C-degron in HEK293T cells with or without shPirh2 treatment. **c** Stability comparison of GFP-fused Ala-3 substitution C-degron to Ala6/C-degron by GPS assay. **d** Stability analysis of GFP-fused Ala-3 substitution C-degron in HEK293T cells with or without shPirh2 treatment. **e** Stability analysis of GFP-fused A-I-A-A C-degron in Pirh2 knockdown (shPirh2) cells ectopically expressing WT Pirh2 or the indicated mutants. **f** Western blot analysis of HA-tagged WT and mutant Pirh2 expression in A-I-A-A C-degron GPS-reporter cell lines. Representative images, $n$ = 3. **g** Stability analysis of GFP-fused S-A-A-A C-degrons in Pirh2 knockdown (shPirh2) cells ectopically expressing WT Pirh2 or the indicated mutants. **h** Western blot analysis of HA-tagged WT and mutant Pirh2 expression in S-A-A-A C-degron GPS-reporter cell lines. Representative images, $n$ = 3. Source data are provided as a Source Data file. All FACS sequential gating images are provided in Supplementary Fig. 9.

participates in the Ala6/C-degron recognition by providing multiple hydrogen bonds and hydrophobic interactions. Structural superposition shows that the E2-binding site is located opposite to the Ala6/C-degron binding surface (Supplementary Fig. 8). This hints that the functions of the RING domain should not be limited to its E2-binding and catalytic activity, but could also be extended to the recognition of specific substrates.

Our studies show that Pirh2 can recognize an Ala-rich C-degron with a variable residue at the −3 or −4 position, implying that Pirh2 probably targets more endogenous substrates containing this

non-strict polyAla motif for ubiquitination, which remains to be explored. Coincidentally, in the bacterial ribosome rescue system, an SsrA tag ending with an A-L-A-A motif to be appended to the nascent polypeptide by tmRNA and acts as a C-degron targeted by ClpXP for proteolysis[33,61]. This A-L-A-A motif is a potential C-degron recognized by human Pirh2, thus at least in part, underscoring the evolutionary convergence of Ala tails in protein quality control.

## Methods

### Cloning, protein expression, and purification

The DNA fragments corresponding to human full-length Pirh2 (residues 1–261), Pirh2 NTD + RING (residues 1–189), Pirh2 NTD (residues 1–137), Pirh2 RING + CTD (residues 138–261), Pirh2 RING (residues 138–189) and Pirh2 CTD (residues 187–261) were cloned by seamless cloning into the pET28-MKH8SUMO vector (Addgene Plasmid #79526), which contains an 8× His-SUMO tag followed by a Tobacco etch virus (TEV) cleavage site. The DNA fragment of Pirh2 (residues 15–191)-linker (GSGSG)-polyAla (AAAAAA) was cloned into the pGEX vector, which contains a GST tag followed by a TEV cleavage site. The recombinant plasmids were transformed into *E. coli* BL21 (DE3) and the expression of the target proteins was induced with 0.2 mM isopropyl β-D-1-thiogalactopyranoside (IPTG) supplemented with 100 μM ZnCl₂ at 18 °C overnight. Cells were harvested by centrifugation and resuspended in lysis buffer (20 mM Tris-HCl pH 8.5, 400 mM NaCl, 5% v/v glycerol, 10 μM ZnCl₂ and 2 mM β-mercaptoethanol). The cells were lysed by sonication on ice and the lysates were clarified by centrifugation at 14,000 rpm for 40 min at 4 °C. The lysate containing GST tagged Pirh2-linker-polyAla was collected to incubate with GST beads for 1 h, and nonspecifically bound proteins were washed out by lysis buffer. The GST tag was proteolytically cleaved by adding the TEV protease to the GST beads overnight at 4 °C and the TEV protease was removed by passing through the Ni-NTA column. The other supernatants were collected to incubate with Ni-NTA beads for 1 h at 4 °C. Nonspecifically bound proteins were washed out of Ni-NTA beads by washing buffer (20 mM Tris-HCl pH 8.5, 400 mM NaCl and 25 mM imidazole pH 8.5) and the fusion proteins were eluted by elution buffer (20 mM Tris-HCl pH 8.5, 400 mM NaCl and 300 mM imidazole pH 8.5). The SUMO tag was cleaved by TEV protease with a molar ration of 1: 20 in dialysis buffer (20 mM Tris-HCl pH 8.5 and 300 mM NaCl) at 4 °C overnight. The samples were reloaded onto the Ni-NTA column to remove the cleaved SUMO tag and TEV protease. Finally, all of the acquired proteins were further purified by size exclusion chromatography (Superdex 200 Increase 10/300 GL, GE healthcare), which was pre-equilibrated with gel-filtration buffer (20 mM Tris-HCl pH 7.5, 150 mM NaCl, 10 μM ZnCl₂ and 1 mM DTT). Purified proteins were concentrated and stored at −80 °C for later use. Expression and purification of the mutants were performed in the same way as wild-type proteins.

### Protein crystallization

For crystallization assays, proteins were concentrated to 8 mg/mL in 20 mM Tris-HCl pH 7.5, 150 mM NaCl, 10 μM ZnCl₂ and 1 mM DTT buffer. Protein crystallization trials were performed using the sitting-drop vapor diffusion method at 18 °C by mixing 1 μL of protein and 1 μL of reservoir solution. Pirh2-linker-polyAla was crystallized in a buffer containing 0.1 M HEPES pH 7.5, 0.1 M NaCl and 1.6 M ammonium sulfate. The crystals were cryoprotected with the respective reservoir solution supplemented with 20% (v/v) glycerol and flash-frozen in liquid nitrogen.

### Data collection and structure determination

The zinc single-wavelength anomalous dispersion (SAD) data sets were collected at Shanghai Synchrotron Radiation Facility (SSRF) beamline BL18U at wavelength of 1.2828 Å and processed with XDS[62]. Zinc SAD phase determination and automatic model building were carried out using AutoSol and AutoBuild implemented in the PHENIX package[63]. The initial partial model was manually rebuilt by Coot[64], and further refined by PHENIX[63].

Schrodinger software suite (Schrödinger, LLC, New York, NY, 2018.) was used for six-Ala peptide docking to KLHDC10, the protein structure predicted by Alphafold was optimized by using "protein preparation wizard" program. The receptor grid was generated by using "binding site detection" program and "receptor grid generation". The 3D structure for the ligand was prepared in Maestro from the 2D structure and parameterized using the OPLS3 force-field. Rigid-receptor/flexible-ligand docking was performed using the extra-precision mode in Glide. The pose with the highest docking score was exported and was analyzed by PyMOL (https://www.pymol.org/2/).

### Isothermal titration calorimetry

Isothermal titration calorimetry (ITC) measurements were performed at 16 °C using a MicroCal PEAQ-ITC instrument (Malvern Panalytical). Proteins and peptides were prepared in an ITC buffer (20 mM Tris-HCl pH 7.5, 150 mM NaCl and 10 μM Zn²⁺). ITC experiments were performed by titrating 1.5 μL (with the exception of the first injection of 0.5 μL) of peptides (0.6–1.0 mM) into cell containing 180–220 μM proteins, with a spacing time of 90 s and a reference power of 10 μcal s⁻¹. The titration data were analyzed using the one-site binding model from MicroCal PEAQ-ITC Analysis Software version 1.30. All ITC assays were repeated at least three times independently with similar results.

### GST pull-down assay

The sequences encoding Ala6/C-degron (sequence: GSGSGSYKT-SAAAAAA) and its mutants (sequence: GSGSGSYKTSAAAAAT, GSGSGSYKTSAAAATA, GSGSGSYKTSAAATAA, GSGSGSYKTSAATAAA, GSGSGSYKTSATAAAA and GSGSGSYKTSTAAAAA) were appended to the C-terminus of GST, and constructed into pET28-MHL vector, respectively. The GST-fusion peptides were purified using GST and Superdex 200 Increase 10/300 GL (GE healthcare) columns. For GST pull-down experiments, approximately 200 μg of GST-fused Ala6/C-degron, degron mutants or GST alone were incubated with 10–20 μL GST beads followed by incubation with equal amounts of the full-length or various Pirh2 fragments for 1 h at 4 °C. Since the molecular weight of full-length Pirh2 is similar to that of GST-fused Ala6/C-degron, we used SUMO-tagged Pirh2 (SUMO-Pirh2) in this experiment. After washing three times by binding buffer (20 mM Tris-HCl pH 7.5, 150 mM NaCl, 10 μM ZnCl₂ and 1 mM DTT), the pulled-down samples were eluted with 25 mM glutathione (pH 7.5). The results were analyzed by SDS-PAGE followed by Coomassie Blue staining.

### Cell culture and viral transduction

Human HEK293T (ATCC CRL-3216) cell line was maintained at 37 °C and 5% CO₂ in Dulbecco's modified Eagle's medium (BioInd) supplemented with 10% fetal bovine serum (BioInd) and 1% penicillin/streptomycin (BioInd). Recombinant viral particles were amplified in the packaging cell line HEK293T, which was produced through co-transfection of HEK293T cells with pMD2.G, pPAX2 (Addgene) and pHAGE, pLKO.1 shRNA or pCDH constructs using Liposomal Transfection Reagent (Yeasen) as recommended by the manufacturer. Lentiviral particle supernatants were collected after 48 h culturing, purified by passing through a 0.45-μm filter and applied for infection of target cells with the presence of 8 μg/mL polybrene. After 48 h, the infected cells were selected with puromycin (2 μg/mL), hygromycin (200 μg/mL) and/or blasticidin (10 μg/mL) for indicated pHAGE/pLKO.1/pCDH stable clones for 3–4 days before the following experiments.

### Global protein stability (GPS) assay

The oligonucleotide encoding Ala6/C-degron and/or mutants was cloned into GPS vector using seamless cloning method, in which the

expression cassette contains a single promoter translating both DsRed and GFP, with an IRES before GFP. DsRed served as an internal control, whereas GFP was expressed fusion with Ala6/C-degron. When integrated into the genome of cells, DsRed and GFP-fused Ala6/C-degron protein were expressed from the same transcript. Knockdown or overexpression of Pirh2 that regulated Ala6/C-degron degradation were expected to change the abundance of GFP-fused Ala6/C-degron, but not DsRed, resulting in GFP/DsRed ratio change. First, the GPS vectors were packaged into lentivirus that were used to transduce HEK293T cells to obtain GPS-reporter cells by puromycin (2 μg/mL) selection. Then endogenous Pirh2 was knocked down using pLKO.1 shRNA lentivirus, and after that HA-tagged wild-type and mutant Pirh2 were overexpressed in GPS-reporter cells followed by hygromycin (200 μg/mL) and blasticidin (10 μg/mL) selection for 3 days. Finally, the stability of the GFP-fused Ala6/C-degron was determined by measuring the cellular GFP/DsRed ratio through flow cytometry using DsRed as an internal control. Data were collected by LSR Fortressa instrument (Becton Dickinson), and analysis of the GFP to DsRed ratio was done through FlowJo software v.10. Expressions of wild-type and mutant Pirh2 in GPS-reporter cells were examined by western blot with anti-HA (Proteintech, 51064-2-AP) at 1: 1000 dilution and anti-alpha tubulin (Proteintech, 11224-1-AP) at 1: 5000 dilution used as the internal control. The antibody validation assay shows that HA antibody recognized HA-fused Pirh2 not the control sample.

## Western blot

In order to verify GFP/DsRed ratio, western blot was performed. First, load each sample into the respective lanes of the SDS-PAGE gel. Run the gel at 120 V until the gel front reaches the bottom. After electrophoresis, place the filter paper, PVDF membrane and gel in the blot module, and insert the module into electrophoresis chamber, transferring proteins from the gel to the PVDF membrane (blot) for 2 h at 300 mA. Incubate the membrane respectively in block solution for 2 h at room temperature, and primary antibody anti-GFP (Proteintech, 50430-2-AP) at 1: 1000 dilution and anti-DsRed (Chromo-Tek, 6G6) at 1: 5000 dilution overnight at 4 °C, secondary antibody solution for 1 h at room temperature. Finally, incubate the membrane in ECL reagent for 1 min in the dark, and then place the blot on a piece of plastic wrap and image. Anti-GFP and Anti-DsRed antibodies were validated by the manufacturer and do not cross-react with other fluorescent proteins.

To test Pirh2-mediated degradation of the native substrate ending in the A/S-X-A-A motif, the Myc tag was fused to the N-terminus of the full-length ITPA protein and the expression level of Myc-fused ITPA was detected using the primary antibody anti-Myc (Proteintech, 16286-1-AP) at 1: 1000 dilution. The antibody validation assay shows that Myc tag antibody recognized Myc-fused ITPA not the control sample.

## RNA interferences and real-time PCR

For Pirh2 and KLHDC10 knockdown, oligonucleotides encoding the sense and antisense strands of short hairpin RNAs (shRNAs) were synthesized, annealed and cloned into the pLKO.1-hygromycin vector. shPirh2 sequence was: CCGGTATCATGTGTCGTCATCTATGCTCGA GCATAGATGACGACACATGATATTTTTG; ShKLHDC10 sequence was: CCGGGAAGAAACCCAGTCGTATATACTCGAGTATATACGACTGGGTTT CTTCTTTTTG. The shRNA targets the 3′-untranslated region (UTR) of endogenous *Pirh2*. Total RNA was extracted using the RNeasy kit (Beyotime) and reverse-transcribed using the cDNA Synthesis Kit (Yeasen). Quantitative real-time PCR (qPCR) analyses were performed using SYBR Green reagent and ABI 7300-Plus Real-time PCR System (Applied Biosystems), with all reactions performed under the following cycling conditions: beginning cycle at 95 °C, 40 cycles at 95 °C for denaturation, followed with 1 min at 65 °C for annealing and extension, and ending with generation of a melting curve consisting of a single peak to rule out non-specific product and primer dimer formations. Each sample was

repeated three times and measured using the mean Cq value. Gene expressions were calculated following normalization to *GAPDH* levels by comparative Ct (cycle threshold) method. Statistic differences were calculated through a two-way unpaired Student's t-test. The paired primers sequences for qPCR were: qPCR-Pirh2: 5′- GAGACAGCTGGAT GATGAAGTAG-3′ and 5′- TGAACAGTGGATCGTCCATTAC-3′, qPCR-KLHDC10: 5′- CAATGGCTCCCTTTATGTCTTTG-3′ and 5′- GTTGTGTCC ACTCTCTGGTATT-3′.

## Reporting summary

Further information on research design is available in the Nature Portfolio Reporting Summary linked to this article.

## Data availability

The atomic coordinates for the reported structures have been deposited in the Protein Data Bank (PDB) under accession codes 7YNX for Pirh2-polyAla. All study data are included in the article and/or SI Appendix. Source data are provided as a Source Data file.

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

## Acknowledgements
We thank the staff at beamline BL18U of Shanghai Synchrotron Radiation Facility for assistance in X-ray data collection. This work was supported by Shandong Province Special Fund "Frontier Technology and Free Exploration" from Pilot National Laboratory for Marine Science and Technology (Qingdao) (No. 8-01), National Natural Science Foundation of China grants 32271265 (to C.D.), 32071193 (to C.D.), 81974431 (to W.M.), 82173000 (to W.M.) and 82103176 (to X.W.), Haihe Laboratory of Cell Ecosystem Innovation Fund (22HHXBSS00048), National Youth Top-Notch Talent Support Program in China, Tianjin Municipal Science and Technology Commission grant 22JCZDJC00440 (to C.D.), Research Foundation of Tianjin Municipal Education Commission grants 2021ZD036 (to C.D.) and 2022KJ191 (to B.Z.), and Core Facility of Research Center of Basic Medical Sciences at Tianjin Medical University.

## Author contributions
C.D. and Y.Y. conceptualized the project and designed experiments. X.J.Y. and Y.L. cloned the constructs and performed protein expression, purification, crystallization, GST pull-down and ITC assays with the help from Q.Y., X.X.Y., and B.Z. X.J.Y. determined the crystal structure. X.W. carried out the cell culture experiments and functional assays with the help from Y.Z. and C.J. D.C., Q.L., T.L., and W.M. analyzed the data. C.D. wrote the manuscript with critical inputs from all authors.

## Competing interests
The authors declare no competing interests.
