## [Peer Review File · Nature Communications]

Recognition of an Ala-rich C-degron by the E3 ligase Pirh2REVIEWER COMMENTS

Reviewer #1 (Remarks to the Author):

The human genome encodes over 650 E3 ligases, each of which targets a specific set of substrates. To achieve specificity in substrate ubiquitination, E3 ligases utilize a wide variety of mechanisms. For example, it has long been known that destabilizing residues at the N-termini of proteins can serve as degrons recognized by the E3 ligase Ubr1. More recently, it has been discovered that sequences at the C-termini of proteins can also function as degrons, and recent years have witnessed rapid identification of both such C-degrons and their respective E3s. In this manuscript, Wang et al structurally investigate how one such E3 ligase, Pirh2, interacts with its C-degron, a polyAla sequence. The structure reveals that the N-terminal domain and RING domain of Pirh2 together form a groove that accommodates the degron and explains the binding selectivity towards polyAla. Moreover, the authors identify Pirh2 residues implicated in the degron interaction. Overall, the authors provide a nice description of the interaction and do a good job at validating the structural model through biophysical and in vivo studies. However, although the studies are technically well executed and will certainly be of interest to a specialized audience, they do not provide sufficient novelty to justify publication in Nature Communications. In particular, that Pirh2 targets both polyAla/C-degrons and Ala-rich C-degrons was known; and how other C-degrons are recognized has been elucidated at the structural level. The study is also limited in that functional consequences of the reported Pirh2 mutations in RQC have not examined (i.e., what is their physiological relevance?); and the work falls short of utilizing the structural understanding to generate important new knowledge.

Specific comments:

1) Fig.5: The finding that Pirh2 knockdown sufficed to cause major stabilization of the polyAla reporter is surprising, because it has been previously reported that the E3 ligase CRL2(KLHDC2) can act redundantly with Pirh2. Can the authors provide an explanation? And what is effect of knocking down KLHDC2 in their assay?

2) "Based on the specificity profile obtained from ITC assays, we suggest that Pirh2 can recognize substrates ending with an A/S-X-A-A motif".

The authors need to test whether Pirh2 targets endogenous substrates that would these sequence criteria.

3) "On the other hand, it is worth to note that in the mammalian RQC pathway, although alanine is the most frequently used residue in the C-terminal tail, 8 other residues including threonine, glycine, asparagine, tryptophan, glutamine, proline, serine and cysteine could be incorporated into the C-terminal tail".

That amino acids other than alanine can be incorporated in mammalian RQC is possible, not unlike amino acid misincorporation in error during canonical translation. However, whether the error rate in C-terminal tail synthesis is high enough to be physiologically relevant is uncertain. Moreover, the evidence for such alternative amino acid incorporation is not rigorous enough to either make such a statement or imply that Pirh2's somewhat relaxed binding sequence preference is meant to accommodate such presumed diversity of C-terminal tails.

4) The authors need to provide more complete details in methods. For example, what were the exact crystallization conditions? How many titrations were used for the ITC experiment? What binding model was used to model the ITC data? One can guess the answers to the latter questions based on the figures, but this type of information is typically stated in methods nonetheless.

Reviewer #2 (Remarks to the Author):

Review for the manuscript NCOMMS-22-39229 by Dr. Dong and co-authors entitled " Recognition of an Ala-rich C-degron by the E3 ligase Pirh2".

RQC (The ribosome-associated quality-control) pathway degrades aberrant nascent polypeptides arising from ribosome stalling during translation. In mammals, the E3 ligases Pirh2 and KLDHC10 mediate the degradation of aberrant nascent polypeptides by targeting the C-terminal polyalanine degrons (polyAla/C-degrons). In this study, the authors determined the crystal structure of Pirh2 bound to the polyAla/C-degron and measured the affinity of the various peptides to Pirh2 in vitro and GFP-A6 reporter proteins in cells in Pirh2-KD condition. Based on these, the authors propose that Pirh2 recognizes a C-terminal A/S-X-A-A motif for substrate degradation by the narrow groove formed by the N-terminal and RING domains. This study provides the molecular basis underlying polyAla/C-degron recognition by Pirh2 and expands the RQC-mediated degradation pathways. These provide an important aspect in the RQC field; however, more experiments are needed to be published in Nature Communications. A more precise analysis is required to propose the model of how Pirh2 recognizes a C-terminal A/S-X-A-A motif for substrate degradation. The consensus sequence for the Pirh2 recognition mainly based on the in vitro binding to Pirh2 must be validated by the stability measurement of GFP-degron reporters.

Comments:

1. To identify the substrate specificity of Pirh2, the authors performed threonine scanning mutagenesis and the GST pull-down assays and ITC using YKTS-A6 peptides. Since the threonine substitutions of Ala-5 and Ala-6 exhibited almost the same binding affinities as the Ala6/C-degron (Fig. 5b), they concluded that four-Ala degron is sufficient for the high-affinity binding to Pirh2. However, the loss of the binding affinity is not always guaranteed Pirh2-mediate GFP-Aal6/C degradation. Pirh2 mutants (Y23A, L92D, R109A) lost the affinity to Ala6/C peptide but were only partially defective in the degradation of GFP-Aal6/C protein (Fig 4c). Indeed, they propose that the long groove formed by the N-terminal and RING domains accommodates six Ala residues (Fig 3g). Therefore, it is necessary to measure the expression level of GFP-Aal6/C protein using threonine substitutions.
2. Validation of the A-2 position for the Pirh2-mediated degradation (Fig. 5d and 6ab): To compare the binding affinity (measured by ITC) and Pirh2-mediated degradation of GFP-A6 reporter, the ITC of the A4XA peptides must be measured.
3. Validation of the A-3 position for the Pirh2-mediated degradation (Fig. 5e and 6c): To compare the binding affinity (measured by ITC) and Pirh2-mediated degradation of GFP-A3XA2 reporters and the ITC of the A3XA2 peptides must be measured. In addition, the flow-cytometry GFP-A3(P/I/K)A2 in the Pirh2-KO condition indicates that GFP-A3(P/I/K)A2 is destabilized by the Pirh2-independent pathway. Since KLDHC10 shares the substrate-specific with Pirh2, it is crucial to validate the contribution of KLDHC10 in the degradation of the GFP-A3(P/I/K)A2 constructs.
4. Validation of the A-1 position for the Pirh2-mediated degradation (Fig. 5c and 6ab): To validate the contribution of the A-1 position for the Pirh2-mediated degradation, it is necessary to measure the expression levels of GFP-Aal6/C proteins with the threonine substitutions and the binding affinity of A6 peptide with threonine substitutions (by ITC).
5. The mutations of Pirh2 at the interaction residues for A-1(Y23A, L92D) or A-4 position Ala (R109A) showed the defect in the degradation of the GFP-A6 (Fig 4c), GFP-A3IA2 (Fig 6e) and GFP-A2SA3 (Fig 6g). These indicate that these mutations affect the GFP-Aal6/C protein degradation independent of the mutation position in the Aal6/C degron. It is reasonable to measure GFP-A5X protein levels in Pirh2-Y23 or L92D mutants and/or GFP-A3XA2 protein levels in Pirh2-R109A mutant, to estimate the position-specific defects of the Pirh2 mutant proteins.

Minor points:

1. The authors described the Dom34-Hbs1-dependent splitting of the stalled ribosome in the introduction. Please check the doi: 10.1016/j.molcel.2012.03.013. that demonstrated the first in vivo evidence for the role of Dom34-Hbs1 in the splitting of the ribosome stalled at the 3' end of the

truncated mRNA (nonstop mRNA). In addition, the RQC-Trigger (RQT) complex is a major dissociation factor to dissociate the collided ribosomes. The RQT complex recognizes ubiquitinated collided ribosomes as a substrate to initiate the RQC pathway by ribosomal subunit dissociation. I believe it is reasonable to refer to the related publications shown below.

The identification of the RQT complex in yeast, doi:10.1038/s41467-017-00188-1 (2017); The in vitro reconstitution of the RQT-mediated splitting of the collided ribosomes in yeast; doi:10.1038/s41594-020-0393-9 (2020); The identification of the RQT complex in mammals, doi:10.1038/s41598-020-60241-w (2020); The in vitro reconstitution of the RQT-mediated splitting of the collided ribosomes in mammals, doi:10.1016/j.molcel.2020.06.006 (2020).

Reviewer #3 (Remarks to the Author):

The paper entitled "Recognition of an Ala-rich C-degron by the E3 ligase Pirh2" reports the molecular basis underlying Ala/C-degron recognition by the E3 ligase Pirh2. Pirh2 together with KLHDC10 was demonstrated by others to function in RQC-C pathway. In this ribosome quality control pathway, Pirh2 was shown to recognize Ala/C-degron and mediates degradation of the ribosome stalled peptide. However, the substrate specificity and recognition mechanism by Pirh2 remained to be explored. In the current study, the Dong lab determined the crystal structure of Pirh2 in complex with a C-terminal polyalanine degron and reveal the molecular mechanism of Pirh2-mediated recognition of polyAla/C-degron. Using structural biology approach, Dong and colleagues elegantly solved the structure of Pirh2 with Ala/C-degron and mapped the critical residues important for substrate recognition. Complemented with biochemical approaches and cell-based assays the authors concluded that the C-degron consensus is S/A-x-A-A and that interestingly, the RING domain that normally is involved in E2 recruitment, is also involved in mediating the interaction with Ala/C-degron containing substrate. Multiple contact sites between Pirh2 and the Ala/C-degron exists and serve to anchor the peptide in the long and narrow Pirh2 binding groove.

Overall, the manuscript is clear and well written. Experiments are well controlled, and the in vitro and in vivo experimental approaches used defined the mode of C-degron recognition by Pirh2. The presented data is valuable for the researchers in the field of protein degradation. However, I would like to point out some issues as follows to help authors clarify and strengthen their conclusions and discussion:

Major comments:

1. In Fig. 2f the authors present ITC assay results of Pirh2 with variable lengths of Ala/C-degron peptides and concluded that "minimal four-Ala degron confers a high-affinity binding to Pirh2". The usage of variable length of peptides is problematic in this assay for a few reasons:
 - a) Although 6 residues peptide accommodates the long groove, residues downstream might assist in binding as well so 10mer peptide cannot be compared to 6mer peptide. Additional residues need to be added to the shorter peptide to make it a uniform 10mer peptide. 6mer "AAAAAA" can also be used to demonstrate that minimal 6mer peptide can interact in those assays.
 - b) Truncated versions of the artificial peptide might contain residues at critical positions that by chance are not compatible with Pirh2 binding. For example- the truncated 6mer "YKTSAA" that does not interact with Pirh2 contains disfavored "T" in the 4th position compared to the 7mer peptide "YKTSAAA" that has a favorable "S" in the 4th position. Thus, a peptide with 2 terminal Ala can still bind if other residues in critical position are favorable, such as "YKSSAA". Authors need to be careful in drawing conclusions based on one artificial peptide with various length of truncated peptide variants.
2. While the authors claim that the RING domain is auxiliary for substrate binding, none of the mutations made to define key residues of Pirh2 for Ala/C-degron binding (Fig. 4) were done on residues in the RING domain. To show the importance of the RING domain, S155 and Ala159 mutants need to be tested in ITC assays.

3. It is clear that Pirh2 plays an important role in PQC mechanisms. It would be nice if authors discuss how many human proteins have favorable Ala/C-degron and might be native substrates of Pirh2. An experimental evidence similar to TRAPPC11 provided by Thrun et al Mol Cell 2021 would be a nice addition.

4. Discussion should include a comparison between Pirh2 and KLHDC10 in terms of their mode of binding to Ala/C-degron. (KLHDC10 structure can be modeled with Alphafold and compared to KLHDC2 structures to predict how it binds Ala tails). The least would be comparing the Pirh2 groove to Kelch solenoid structure and hence mode of binding.

Minor comments:

1. Page 3: "One such surveillance pathway is ribosome-associated quality control (RQC) that targets the potentially toxic nascent polypeptides produced by defective translation for degradation"

Correct to: "One such surveillance pathway is ribosome-associated quality control (RQC) that targets for degradation the potentially toxic nascent polypeptides produced by defective translation"

2. Page 3: "During this process, the free 40S subunit can be recycled, and the aberrant mRNA is degraded by exoribonuclease and the exosome complex^{10,18}. While the incomplete nascent chains remain attached to the 60S subunit, and therefore are about to undergo further processing via RQC-L or RQC-C pathway^{19, 20}"

Correct to: "During this process, the free 40S subunit can be recycled, and the aberrant mRNA is degraded by exoribonuclease and the exosome complex^{10,18}, while the incomplete nascent chains remain attached to the 60S subunit, and therefore are about to undergo further processing via RQC-L or RQC-C pathway^{19, 20}"

3. Page 3: "If the Listerin activity is limited under certain conditions, the aberrant nascent chain could be proteolytic..."

"proteolytic" correct to "proteolyzed"

4. Page 3: "In which, the NEMF..." -

Replace "In which" with "In RQC-C"

5. Page 5: "To better understand the recognition mode of Ala6/C-degron by Pirh2. We determined the crystal structure of Pirh2 NTD+RING domains bound to the Ala6/C-degron peptide".

Correct to: "To better understand the recognition mode of Ala6/C-degron by Pirh2, we determined the crystal structure of Pirh2 NTD+RING domains bound to the Ala6/C-degron peptide"

Reviewer #4 (Remarks to the Author):

In this study, the authors have used a combination of crystallography, in vitro binding assays, and degradation assays in cells to analyze how the ubiquitin ligase Pirh2 recognizes its C-terminal poly-Ala substrates. The study first maps which domains of Pirh2 are required for poly-Ala interaction. After defining the domain, they crystallize it as an in-line fusion with the poly-Ala substrate peptide. This nicely explains how the substrate is recognized and reveals the surprising result that the RING domain participate intimately in this recognition. The structure is then used for two types of mutagenesis. First, the Pirh2 binding site is mutated to verify the structure in a combination of in vitro Kd measurements and degradation assays in cells. Second, the tolerance of Pirh2 for variant substrates is tested by mutagenesis of the poly-Ala motif, again using in vitro binding assays and degradation assay in cells. The results lead to an understanding of how Pirh2 recognizes substrates as part of the C-degron limb of the RQC pathway. Notably, the mode of recognition seems to be different than what would occur for other substrates such as p53.

The study is of interest to colleagues in the protein quality control field and those studying ubiquitin ligases. The quality of the crystal structure seems to be high (although this is not my area of expertise), and the binding mode is validated rigorously with mutants. I therefore believe the study is of high quality and an important contribution to the fields noted above. As such, I strongly support its publication with only minor revisions to the text.

Minor suggestions:

1. Introduction – The Pelota/Hbs1 mechanism of ribosome splitting probably only operates when the A site of the ribosome is empty such as on a truncated mRNA. Splitting after stalling in the middle of an mRNA seems to involve ribosome collisions (PMID 30293783 and 30609991), 40S ubiquitination by ZNF598 (28065601), and ribosome dissociation by the ASCC (PMID 32579943).

2. Pg. 5 – it is stated that the authors determined “the crystal structure of Pirh2 NTD+RING domains bound to the Ala6/C-degron peptide.”. However, this is not quite correct. If I understood the Methods correctly, the protein that was crystallized was a fusion protein in which the C-terminal peptide is directly linked to the Pirh2 protein. Please state this directly in the text so a reader does not get the false impression that it is a co-crystal with a peptide substrate. One could add a dotted line in Fig. 1a showing where the linker is that attaches to the substrate peptide.

3. Please proofread and adjust grammar throughout prior to publication.

Reviewer #5 (Remarks to the Author):

Excellent body of work and a very easy manuscript to review.

The experiments are well controlled.

The figure set is well presented.

The conclusions drawn are well supported by the data.

Nice balance between a definitive structure, in vitro characterization and in cell validation.

I recommend publication of this nice body of work without need for further experimentation.

All that is needed is some polishing of the grammar.

Response to Reviewer #1

Comments:

The human genome encodes over 650 E3 ligases, each of which targets a specific set of substrates. To achieve specificity in substrate ubiquitination, E3 ligases utilize a wide variety of mechanisms. For example, it has long been known that destabilizing residues at the N-termini of proteins can serve as degrons recognized by the E3 ligase Ubr1. More recently, it has been discovered that sequences at the C-termini of proteins can also function as degrons, and recent years have witnessed rapid identification of both such C-degrons and their respective E3s. In this manuscript, Wang et al structurally investigate how one such E3 ligase, Pirh2, interacts with its C-degron, a polyAla sequence. The structure reveals that the N-terminal domain and RING domain of Pirh2 together form a groove that accommodates the degron and explains the binding selectivity towards polyAla. Moreover, the authors identify Pirh2 residues implicated in the degron interaction. Overall, the authors provide a nice description of the interaction and do a good job at validating the structural model through biophysical and in vivo studies. However, although the studies are technically well executed and will certainly be of interest to a specialized audience, they do not provide sufficient novelty to justify publication in Nature Communications. In particular, that Pirh2 targets both polyAla/C-degrons and Ala-rich C-degrons was known; and how other C-degrons are recognized has been elucidated at the structural level. The study is also limited in that functional consequences of the reported Pirh2 mutations in RQC have not been examined (i.e., what is their physiological relevance?); and the work falls short of utilizing the structural understanding to generate important new knowledge.

Response:

Thank you very much for your constructive suggestions on our manuscript, which help us to carefully revise it and provide experiments. To examine the functional consequences of the reported Pirh2 mutations in RQC, we generated the DsRed-IRES-GFP NS (non-stop) construct according to the reported RQC-reporter system¹. The HEK293T cells stably expressing the GFP NS reporter were treated with shRNA targeting Pirh2 and with the Neddylation inhibitor MLN4924 (1 μ M for 6 h) that inhibits KLHDC10 activity. Western blot analysis showed that Pirh2 knockdown stabilized the GFP NS (Fig. R1a). However, overexpression of wild-type Pirh2, but not the C-degron-binding-deficient mutants including Y23A, L92D and R109A, significantly promoted the degradation of the GFP NS (Fig. R1b), suggesting that the residues of Pirh2-mediated polyAla/C-degron recognition play an essential role in the RQC-mediated degradation of GFP NS.

Fig. R1 | Western blotting analysis of GFP NS stability. **a** Stability analysis of GFP NS with Pirh2 KD in HEK293T cells by western blotting. **b** Stability analysis of GFP NS with over-expressed WT and indicated mutant Pirh2 proteins in HEK293T cells by western blotting. Representative images, n=3.

Specific comments:

1) Fig.5: The finding that Pirh2 knockdown sufficed to cause major stabilization of the polyAla reporter is surprising, because it has been previously reported that the E3 ligase CRL2(KLHDC2) can act redundantly with Pirh2. Can the authors provide an explanation? And what is effect of knocking down KLHDC2 in their assay?

Response:

We thank the reviewer for pointing out this. We have repeated the GPS experiment and found that knocking down Pirh2 or KLHDC10 individually does indeed result in a major stabilization of the polyAla reporter under our experimental conditions (Fig. R2). A likely interpretation of these results is that we performed the GPS experiment in HEK293T cells but not in the previously reported RPE1 cells, and we knocked down Pirh2 or KLHDC10 by shRNA instead of the previously reported siRNA.

In addition, as per your suggestion we knocked down both Pirh2 and KLHDC10 simultaneously causes greatly increased GFP-Ala6 levels (Fig. R2), which is consistent with previous results¹.

Fig. R2 | Stability analysis of GFP-Ala6 degron upon Pirh2 and KLHDC10 single or double knockdown in HEK293T cells by flow cytometry. The ratio of GFP/DsRed was analyzed by flow cytometry.

2) “Based on the specificity profile obtained from ITC assays, we suggest that Pirh2 can recognize substrates ending with an A/S-X-A-A motif”.

The authors need to test whether Pirh2 targets endogenous substrates that would these sequence criteria.

Response:

We thank the reviewer for this suggestion. We carried out A/S-X-A-A motif search against the UniProt database, and found that the ITPA (inosine triphosphate pyrophosphatase) protein ending in SLAA (~LQEYFGSLAA) is a potential substrate of Pirh2.

Indeed, Myc-fused ITPA levels were increased upon Pirh2 knockdown by shRNA. Furthermore, overexpression of wild-type Pirh2, but not the C-degion-binding-deficient mutants including Y23A, L92D or R109A, caused a reduction in ITPA levels.

We have added these data in the revised manuscript (Supplementary Fig. 5).

Supplementary Fig. 5 | Stability analysis of full length ITPA (inosine triphosphate pyrophosphatase) by western blotting. a Stability analysis of Myc-tagged full-length ITPA upon Pirh2 knockdown in HEK293T cells by western blotting. **b**, Stability analysis of Myc-tagged full-length ITPA with over-expressed HA-tagged Pirh2 (WT or mutant) in HEK293T cells by western blotting.

3) “On the other hand, it is worth to note that in the mammalian RQC pathway, although alanine is the most frequently used residue in the C-terminal tail, 8 other residues including threonine, glycine, asparagine, tryptophan, glutamine, proline, serine and cysteine could be incorporated into the C-terminal tail”.

That amino acids other than alanine can be incorporated in mammalian RQC is possible, not unlike amino acid misincorporation in error during canonical translation. However, whether the error rate in C-terminal tail synthesis is high enough to be physiologically relevant is uncertain. Moreover, the evidence for such alternative amino acid incorporation is not rigorous enough to either make such a statement or imply that Pirh2’s somewhat relaxed binding sequence preference is meant to accommodate such presumed diversity of C-terminal tails.

Response:

Thank you for this critical comment, we agree and have removed this discussion part from the revised manuscript.

4) The authors need to provide more complete details in methods. For example, what were the exact crystallization conditions? How many titrations were used for the ITC experiment? What binding model was used to model the ITC data? One can guess the answers to the latter questions based on the figures, but this type of information is typically stated in methods nonetheless.

Response:

Thank you for your kind suggestions. We have added the method details of the ITC experiment, as well as the other experiments in the revised manuscript.

Response to Reviewer #2

Comments:

Review for the manuscript NCOMMS-22-39229 by Dr. Dong and co-authors entitled " Recognition of an Ala-rich C-degron by the E3 ligase Pirh2".

RQC (The ribosome-associated quality-control) pathway degrades aberrant nascent polypeptides arising from ribosome stalling during translation. In mammals, the E3 ligases Pirh2 and KLDHC10 mediate the degradation of aberrant nascent polypeptides by targeting the C-terminal polyalanine degrons (polyAla/C-degrons). In this study, the authors determined the crystal structure of Pirh2 bound to the polyAla/C-degron and measured the affinity of the various peptides to Pirh2 in vitro and GFP-A6 reporter proteins in cells in Pirh2-KD condition. Based on these, the authors propose that Pirh2 recognizes a C-terminal A/S-X-A-A motif for substrate degradation by the narrow groove formed by the N-terminal and RING domains. This study provides the molecular basis underlying polyAla/C-degron recognition by Pirh2 and expands the RQC-mediated degradation pathways. These provide an important aspect in the RQC field; however, more experiments are needed to be published in Nature Communications. A more precise analysis is required to propose the model of how Pirh2 recognizes a C-terminal A/S-X-A-A motif for substrate degradation. The consensus sequence for the Pirh2 recognition mainly based on the in vitro binding to Pirh2 must be validated by the stability measurement of GFP-degron reporters.

Response:

Thank you very much for your positive comments and constructive suggestions on our manuscript. We have provided the experiments you requested to strengthen the manuscript and address these concerns as follows.

Major comments:

1. To identify the substrate specificity of Pirh2, the authors performed threonine scanning mutagenesis and the GST pull-down assays and ITC using YKTS-A6

peptides. Since the threonine substitutions of Ala-5 and Ala-6 exhibited almost the same binding affinities as the Ala6/C-degron (Fig. 5b), they concluded that four-Ala degron is sufficient for the high-affinity binding to Pirh2. However, the loss of the binding affinity is not always guaranteed Pirh2-mediate GFP-Aal6/C degradation. Pirh2 mutants (Y23A, L92D, R109A) lost the affinity to Ala6/C peptide but were only partially defective in the degradation of GFP-Aal6/C protein (Fig 4c). Indeed, they propose that the long groove formed by the N-terminal and RING domains accommodates six Ala residues (Fig 3g). Therefore, it is necessary to measure the expression level of GFP-Aal6/C protein using threonine substitutions.

Response:

We thanks for your nice suggestion. As per your suggestion, we measured the expression levels of GFP-Aal6/C protein using threonine substitutions. To do this, we examined the effects of Pirh2 on the stability of GFP-fused Ala6/C-degron with threonine substitutions in cells in which KLHDC10, another E3 ligase that can act redundantly with Pirh2 in RQC, had been knocked down by shRNA. Our GPS assays showed that the threonine substitutions of Ala-1, Ala-2 and Ala-4 led to a major stabilization of GFP, whereas the substitutions of Ala-3, Ala-5 and Ala-6 had almost the same behavior as the wild-type Ala6/C-degron, which is consistent with our in vitro results.

We have added this data in the revised manuscript (Supplementary Fig. 2).

Supplementary Fig. 2 | Stability analysis of different substitutions of Ala6/C-degron. **a** The relative mRNA levels of KLHDC10 in shNT and shKLHDC10 cells. Error bars indicate S.E.M. of three biological replicates. P value was determined using unpaired two-tailed Student's t-tests; n=3 biologically independent samples. **b** The stability analysis of GFP-fused the indicated sequences in HEK293T cells with KLHDC10 knocked down.

2. Validation of the A-2 position for the Pirh2-mediated degradation (Fig. 5d and 6ab): To compare the binding affinity (measured by ITC) and Pirh2-mediated degradation of GFP-A6 reporter, the ITC of the A4XA peptides must be measured.

Response:

Thank you very much for your suggestions. We have supplied the ITC titration results of A4XA peptide, the results showed that the A4XA peptide including A4VA, A4IA and A4SA, caused a significant decrease in the binding affinity compared to the A6 peptide (Fig. R3).

We have added this data in the revised manuscript (Supplementary Fig. 3a).

Fig. R3 | ITC fitting curves of Pirh2 titrated by the peptide with substitutions at position -2.

3. Validation of the A-3 position for the Pirh2-mediated degradation (Fig. 5e and 6c): To compare the binding affinity (measured by ITC) and Pirh2-mediated degradation of GFP-A3XA2 reporters and the ITC of the A3XA2 peptides must be measured. In addition, the flow-cytometry GFP-A3(P/I/K)A2 in the Pirh2-KO condition indicates that GFP-A3(P/I/K)A2 is destabilized by the Pirh2-independent pathway. Since KLDHC10 shares the substrate-specific with Pirh2, it is crucial to validate the contribution of KLDHC10 in the degradation of the GFP-A3(P/I/K)A2 constructs.

Response:

We thanks for your suggestion. We have measured the binding affinity of A3XA2 (such as A3PA2, A3IA2 and A3KA2) by ITC, the results showed that the A3XA2 peptides had a robust binding to Pirh2 (Fig. R4). We have added this data in the revised manuscript (Supplementary Fig. 3b).

Fig. R4 | ITC fitting curves of Pirh2 titrated by the peptide with substitutions at position -3.

Furthermore, we followed the reviewer's suggestion and performed GPS assays using GFP-A3IA2 construct, the results showed that GFP-A3IA2 was also dramatically destabilized upon KLHDC10 knockdown, while Pirh2 and KLHDC10

double knockdown had stronger effect compared to Pirh2 or KLHDC10 silencing alone (Fig. R5), suggesting that LHDC10 can act redundantly with Pirh2 in promoting the degradation of similar polyAla/C-degron motif.

Fig. R5. Stability analysis of GFP-A3IA2 upon Pirh2 and KLHDC10 single or double knockdown in HEK293T cells.

4. Validation of the A-1 position for the Pirh2-mediated degradation (Fig. 5c and 6ab): To validate the contribution of the A-1 position for the Pirh2-mediated degradation, it is necessary to measure the expression levels of GFP-Aal6/C proteins with the threonine substitutions and the binding affinity of A6 peptide with threonine substitutions (by ITC).

Response:

Thank you for your suggestion. We have measured the expression levels of GFP-Aal6/C proteins with the threonine substitutions of A-1, our GPS assays showed that the threonine substitutions of A-1 caused a significant stabilization of GFP compared to Ala6/C-degron (Supplementary Figure 2). In parallel, ITC results indicated that the threonine substitutions of A-1 completely abolished the degron-binding activity (Fig. 5b).

Supplementary Fig. 2 | Stability analysis of different substitutions of Ala6/C-degron. a The relative mRNA levels of KLHDC10 in shNT and shKLHDC10 cells. Error bars indicate S.E.M. of three biological replicates. P values were determined using unpaired two-tailed Student's t-tests; n=3 biologically independent samples. **b** Flow cytometry analysis of GFP-fused the indicated sequences with KLHDC10 knocked down.

Fig. 5b | ITC fitting curves of Pirh2 (NTD+RING) titrated by a series of peptides with position substitutions. The corresponding peptide sequences and binding affinities (K_D) are indicated. NB, no detectable binding under our experimental conditions.

5. The mutations of Pirh2 at the interaction residues for A-1(Y23A, L92D) or A-4 position Ala (R109A) showed the defect in the degradation of the GFP-A6 (Fig 4c), GFP-A3IA2 (Fig 6e) and GFP-A2SA3 (Fig 6g). These indicate that these mutations affect the GFP-Aal6/C protein degradation independent of the mutation position in the Aal6/C degron. It is reasonable to measure GFP-A5X protein levels in Pirh2-Y23 or L92D mutants and/or GFP-A3XA2 protein levels in Pirh2-R109A mutant, to estimate the position-specific defects of the Pirh2 mutant proteins.

Response:

Thank you for your suggestion. As per your suggestion, we have measured GFP-A5T protein levels in Pirh2-Y23 or L92D mutants, and GFP-A2SA3 and GFP-A2TA3 (substitution of the position -4, if I understood correctly) protein levels in Pirh2-R109A mutant. Our GPS assays showed that in the Pirh2 and KLHDC10 double knockdown cells, the GFP-A5T protein was stabilized upon overexpression of either the A-1 deficient mutant Y23A/L92D or WT Pirh2 (Fig. R6a and R6b). Likewise, GFP-A2TA3 protein was stabilized upon overexpression of the A-4 deficient mutant R109A or WT Pirh2 (Fig. R6c). Furthermore, overexpression of the A-4 deficient mutant R109A, like other Pirh2 mutants, resulted in stabilization of GFP-A2SA3 protein (Fig. R6d). Therefore, it is likely to that the Pirh2 mutations affect the GFP-Aal6/C protein degradation independent of the mutation position in the Aal6/C degron, probably due to the fact that the degron-interacting residues of Pirh2 mainly bind to the backbone of the polyAla/C-degron, and provide the strict spatial conformation.

Fig. R6 | Stability analysis of WT or mutant Ala6/C-degron in HEK293T cells with stably expressing Pirh2 (WT or mutant) upon Pirh2 and KLHDC10 double knockdown. a Stability assay of A5T degron in Pirh2 WT and Y23A mutant; **b** Stability assay of A5T degron in Pirh2 WT and L92D mutant; **c** Stability assay of A2TA3 degron in Pirh2 WT and R109A mutant; **d** Stability assay of A2SA3 degron in Pirh2 WT and R109A mutant.

Minor points:

1. The authors described the Dom34-Hbs1-dependent splitting of the stalled ribosome in the introduction. Please check the doi: 10.1016/j.molcel.2012.03.013. that demonstrated the first in vivo evidence for the role of Dom34-Hbs1 in the splitting of the ribosome stalled at the 3' end of the truncated mRNA (nonstop mRNA). In addition, the RQC-Trigger (RQT) complex is a major dissociation factor to dissociate the collided ribosomes. The RQT complex recognizes ubiquitinated collided ribosomes as a substrate to initiate the RQC pathway by ribosomal subunit dissociation. I believe it is reasonable to refer to the related publications shown below.

The identification of the RQT complex in yeast, doi:10.1038/s41467-017-00188-1 (2017); The in vitro reconstitution of the RQT-mediated splitting of the collided ribosomes in yeast; doi:10.1038/s41594-020-0393-9 (2020); The identification of the RQT complex in mammals, doi:10.1038/s41598-020-60241-w (2020); The in vitro reconstitution of the RQT-mediated splitting of the collided ribosomes in mammals, doi:10.1016/j.molcel.2020.06.006 (2020).

Response:

Thank you very much for your suggestions. We have cited these right papers in the revised manuscript.

Response to Reviewer #3

Comments:

The paper entitled “Recognition of an Ala-rich C-degron by the E3 ligase Pirh2” reports the molecular basis underlying Ala/C-degron recognition by the E3 ligase Pirh2. Pirh2 together with KLHDC10 was demonstrated by others to function in RQC-C pathway. In this ribosome quality control pathway, Pirh2 was shown to recognize Ala/C-degron and mediates degradation of the ribosome stalled peptide. However, the substrate specificity and recognition mechanism by Pirh2 remained to be explored. In the current study, the Dong lab determined the crystal structure of Pirh2 in complex with a C-terminal polyalanine degron and reveal the molecular mechanism of Pirh2-mediated recognition of polyAla/C-degron. Using structural biology approach, Dong and colleagues elegantly solved the structure of Pirh2 with Ala/C-degron and mapped the critical residues important for substrate recognition. Complemented with biochemical approaches and cell-based assays the authors concluded that the C-degron consensus is S/A-x-A-A and that interestingly, the RING domain that normally is involved in E2 recruitment, is also involved in mediating the interaction with Ala/C-degron containing substrate. Multiple contact sites between Pirh2 and the Ala/C-degron exists and serve to anchor the peptide in the long and narrow Pirh2 binding groove.

Overall, the manuscript is clear and well written. Experiments are well controlled, and the *in vitro* and *in vivo* experimental approaches used defined the mode of C-degron recognition by Pirh2.

The presented data is valuable for the researchers in the field of protein degradation. However, I would like to point out some issues as follows to help authors clarify and strengthen their conclusions and discussion:

Major comments:

1. In Fig. 2f the authors present ITC assay results of Pirh2 with variable lengths of Ala/C-degron peptides and concluded that “minimal four-Ala degron confers a high-affinity binding to Pirh2”. The usage of variable length of peptides is problematic in this assay for a few reasons:

a) Although 6 residues peptide accommodates the long groove, residues downstream might assist in binding as well so 10mer peptide cannot be compared to 6mer peptide. Additional residues need to be added to the shorter peptide to make it a uniform 10mer peptide. 6mer “AAAAAA” can also to be used to demonstrate that minimal 6mer peptide can interact in those assays.

b) Truncated versions of the artificial peptide might contain residues at critical positions that by chance are not compatible with Pirh2 binding. For example- the truncated 6mer “YKTSAA” that does not interact with Pirh2 contains disfavored “T” in the 4th position compared to the 7mer peptide “YKTSAAA” that has a favorable “S” in the 4th position. Thus, a peptide with 2 terminal Ala can still bind if other residues in critical position are favorable, such as “YKSSAA”.

Authors need to be careful in drawing conclusions based on one artificial peptide with

various length of truncated peptide variants.

Response:

Thank you very much for this constructive suggestion. We agree, and to rule out the interference of other residues in the longer peptide, we synthesized 6mer “AAAAAA” peptides with various lengths of truncated variants. Our ITC results indicated that the peptide containing four to six Ala residues retains binding activities, while the three-Ala or two-Ala peptide resulted in significant loss of Pirh2 binding (Fig. R7). Therefore, we conclude that the minimal four-Ala degron is sufficient to bind Pirh2, which is in agreement with previous data that at least four-Ala degron can be efficiently degraded by Pirh2¹.

We have added these data in the revised manuscript (Fig. 2f).

Fig. R7 | ITC fitting curves of Pirh2 (NTD+RING) with variable lengths of Ala6/C-degron peptides. The corresponding peptide lengths, sequences and binding affinities (K_D) are indicated. NB, no detectable binding under our experimental conditions.

2. While the authors claim that the RING domain is auxiliary for substrate binding, none of the mutations made to define key residues of Pirh2 for Ala/C-degron binding (Fig. 4) were done on residues in the RING domain. To show the importance of the RING domain, S155 and Ala159 mutants need to be tested in ITC assays.

Response:

Thank you for your nice suggestions, we generated the mutants S155D and A159D in the RING domain and detected their binding with Ala6/C-degron by ITC experiment. The results showed that the mutants S155D and A159D decreased the binding affinity by 9- and 14-fold, respectively (Fig. R8), suggesting that the RING domain also plays an important role in the Ala6/C-degron binding.

We have added these data in the revised manuscript (Fig. 4a).

Fig. R8 | ITC fitting curves of wild-type (WT) and mutant Pirh2 titrated with the Ala6/C-degron peptide. a ITC fitting curves of wild-type (WT) and S155D mutant titrated with the Ala6/C-degron peptide. **b** ITC fitting curves of wild-type (WT) and A159D mutant titrated with the Ala6/C-degron peptide. NB, no detectable binding under our experimental conditions.

3. It is clear that Pirh2 plays an important role in PQC mechanisms. It would be nice if authors discuss how many human proteins have favorable Ala/C-degron and might be native substrates of Pirh2. An experimental evidence similar to TRAPPC11 provided by Thrun et al Mol Cell 2021 would be a nice addition.

Response:

We thank the reviewer for this suggestion. We carried out A/S-X-A-A motif search against the UniProt database, and found that the ITPA (inosine triphosphate pyrophosphatase) protein ending in SLAA (~LQEYFGSLAA) is a potential substrate of Pirh2.

Indeed, Myc-fused ITPA levels were increased upon Pirh2 knockdown by shRNA. Furthermore, overexpression of wild-type Pirh2, but not the C-degron-binding-deficient mutants including Y23A, L92D or R109A, caused a reduction in ITPA levels.

We have added these data in the revised manuscript (Supplementary Fig. 5).

Supplementary Fig. 5 | Stability analysis of full length ITPA (inosine triphosphate pyrophosphatase) by western blotting. a Stability analysis of Myc-tagged full-length ITPA upon Pirh2 knockdown in HEK293T cells by western blotting. **b**, Stability analysis of Myc-tagged full-length ITPA with over-expressed HA-tagged Pirh2 (WT or mutant) in HEK293T cells by western blotting. Representative images, n=3. Uncropped western blot images are provided in Source data 1.

4. Discussion should include a comparison between Pirh2 and KLHDC10 in terms of their mode of binding to Ala/C-degron. (KLHDC10 structure can be modeled with Alphafold and compared to KLHDC2 structures to predict how it binds Ala tails). The least would be comparing the Pirh2 groove to Kelch solenoid structure and hence mode of binding.

Response:

As per your suggestion, we have added a discussion on the comparison between Pirh2 and KLHDC10 in terms of their mode of binding to Ala/C-degron in the manuscript. It now reads: “In addition to Pirh2, the ployAla/C-degron is also sensed by the E3 ligase KLHDC10 , which contains a Kelch repeat domain for substrate binding. We predicted the binding mode of KLHDC10 to the six-Ala tail by means of the structure predicted by Alphafold and molecular docking. The modeled ployAla/C-degron is embedded in a positively charged pocket, similar to that of other typical C-degron binding modes²⁻⁴, and is shown here (Supplementary Fig. 6). Noteworthy, KLHDC10 utilizes a deep and narrow pocket to engage the six-Ala tail with a solenoidal conformation similar to the reported Gly/C-degron-bound KLHDC2 structure. However, Pirh2 uses a long groove to recognize the linear six-Ala tail. Therefore, Pirh2 and KLHDC10 may tend to recognize their dedicated polyAla tail in different conformations”.

Supplementary Fig. 6 | The electrostatic potential surface of the Ala6/C-degron binding pocket in KLHDC10 (red, negative; blue, positive). The six-Ala peptide was docked into the Alphafold-predicted KLHDC10 structure using the Schrodinger software suite.

Minor comments:

1. Page 3: “One such surveillance pathway is ribosome-associated quality control (RQC) that targets the potentially toxic nascent polypeptides produced by defective translation for degradation”

Correct to: “One such surveillance pathway is ribosome-associated quality control (RQC) that targets for degradation the potentially toxic nascent polypeptides produced by defective translation”

Response:

According to your kind suggestion, we have revised this sentence in the revised manuscript.

2. Page 3: “During this process, the free 40S subunit can be recycled, and the aberrant mRNA is degraded by exoribonuclease and the exosome complex^{10,18}. While the incomplete nascent chains remain attached to the 60S subunit, and therefore are about to undergo further processing via RQC-L or RQC-C pathway^{19, 20}”

Correct to: “During this process, the free 40S subunit can be recycled, and the aberrant mRNA is degraded by exoribonuclease and the exosome complex^{10,18}, while the incomplete nascent chains remain attached to the 60S subunit, and therefore are about to undergo further processing via RQC-L or RQC-C pathway^{19, 20}”

Response:

According to your kind suggestion, we have revised this sentence as the reviewer suggested.

3. Page 3: “If the Listerin activity is limited under certain conditions, the aberrant nascent chain could be proteolytic...”.
“proteolytic” correct to “proteolyzed”

Response:

According to your kind suggestion, we have revised this sentence as the reviewer suggested.

4. Page 3: “In which, the NEMF...” -
Replace “In which” with “In RQC-C”

Response:

According to your kind suggestion, we have revised this sentence as the reviewer suggested.

5. Page 5: “To better understand the recognition mode of Ala6/C-degron by Pirh2. We determined the crystal structure of Pirh2 NTD+RING domains bound to the Ala6/C-degron peptide”.

Correct to: “To better understand the recognition mode of Ala6/C-degron by Pirh2, we determined the crystal structure of Pirh2 NTD+RING domains bound to the Ala6/C-degron peptide”

Response:

According to your kind suggestion, we have revised this sentence as the reviewer suggested.

Response to Reviewer #4

In this study, the authors have used a combination of crystallography, in vitro binding assays, and degradation assays in cells to analyze how the ubiquitin ligase Pirh2 recognizes its C-terminal poly-Ala substrates. The study first maps which domains of Pirh2 are required for poly-Ala interaction. After defining the domain, they crystallize it as an in-line fusion with the poly-Ala substrate peptide. This nicely explains how the substrate is recognized and reveals the surprising result that the RING domain participate intimately in this recognition. The structure is then used for two types of mutagenesis. First, the Pirh2 binding site is mutated to verify the structure in a combination of in vitro K_d measurements and degradation assays in cells. Second, the tolerance of Pirh2 for variant substrates is tested by mutagenesis of the poly-Ala motif, again using in vitro binding assays and degradation assay in cells. The results lead to an understanding of how Pirh2 recognizes substrates as part of the C-degron limb of the RQC pathway. Notably, the mode of recognition seems to be different than what would occur for other substrates such as p53.

The study is of interest to colleagues in the protein quality control field and those studying ubiquitin ligases. The quality of the crystal structure seems to be high (although this is not my area of expertise), and the binding mode is validated rigorously with mutants. I therefore believe the study is of high quality and an important contribution to the fields noted above. As such, I strongly support its publication with only minor revisions to the text.

Minor suggestions:

1. Introduction – The Pelota/Hbs1 mechanism of ribosome splitting probably only operates when the A site of the ribosome is empty such as on a truncated mRNA. Splitting after stalling in the middle of an mRNA seems to involve ribosome collisions (PMID 30293783 and 30609991), 40S ubiquitination by ZNF598 (28065601), and ribosome dissociation by the ASCC (PMID 32579943).

Response:

Thank you for your kind suggestions, we have added the detailed statements and cited the related paper in the revised manuscript.

6. Pg. 5 – it is stated that the authors determined “the crystal structure of Pirh2 NTD+RING domains bound to the Ala6/C-degron peptide.”. However, this is not quite correct. If I understood the Methods correctly, the protein that was crystallized was a fusion protein in which the C-terminal peptide is directly linked to the Pirh2 protein. Please state this directly in the text so a reader does not get the false impression that it is a co-crystal with a peptide substrate. One could add a dotted line in Fig. 1a showing where the linker is that attaches to the substrate peptide.

Response:

Thank you very much for your suggestions, we have revised the statement in the

revised manuscript. The dotted line have also been added in Fig. 1a to show the linker.

3. Please proofread and adjust grammar throughout prior to publication.

Response:

Thank you very much for your suggestions, we have carefully proofread and adjust grammar throughout our manuscript.

Response to Reviewer #5

Excellent body of work and a very easy manuscript to review.

The experiments are well controlled.

The figure set is well presented.

The conclusions drawn are well supported by the data.

Nice balance between a definitive structure, in vitro characterization and in cell validation.

I recommend publication of this nice body of work without need for further experimentation.

All that is needed is some polishing of the grammar.

Response:

Thank you very much for your positive comments and suggestions. As the reviewer suggested, we have polished the grammar throughout the manuscript carefully.

References

- 1 Thrun, A. *et al.* Convergence of mammalian RQC and C-end rule proteolytic pathways via alanine tailing. *Molecular cell*, doi:10.1016/j.molcel.2021.03.004 (2021).
- 2 Yan, X. *et al.* Molecular basis for ubiquitin ligase CRL2(FEM1C)-mediated recognition of C-degron. *Nature chemical biology*, doi:10.1038/s41589-020-00703-4 (2021).
- 3 Rusnac, D. V. *et al.* Recognition of the Diglycine C-End Degron by CRL2(KLHDC2) Ubiquitin Ligase. *Molecular cell* **72**, 813-822 e814, doi:10.1016/j.molcel.2018.10.021 (2018).
- 4 Ru, Y. *et al.* C-terminal glutamine acts as a C-degron targeted by E3 ubiquitin ligase TRIM7. *Proceedings of the National Academy of Sciences of the United States of America* **119**, e2203218119, doi:10.1073/pnas.2203218119 (2022).

REVIEWERS' COMMENTS

Reviewer #2 (Remarks to the Author):

Review for the manuscript NCOMMS-22-39229A by Dr. Dong and co-authors entitled " Recognition of an Ala-rich C-degron by the E3 ligase Pirh2"

The authors have addressed most of my previous concerns. I support the publication of manuscript in Nature Communications.

Reviewer #3 (Remarks to the Author):

The major experimental concerns I and other reviewers had were addressed with new experiments and text editing, which is appreciated. As such the study is suitable for publication.

Response to Reviewer #2

Comments:

The authors have addressed most of my previous concerns. I support the publication of manuscript in Nature Communications.

Response:

Thank you very much for reviewing our manuscript and for your constructive comments, which have helped us a lot to improve the manuscript.

Response to Reviewer #3

Comments:

The major experimental concerns I and other reviewers had were addressed with new experiments and text editing, which is appreciated. As such the study is suitable for publication.

Response:

Thank you very much for reviewing our manuscript and for your constructive comments, which have helped us a lot to improve the manuscript.